Resource

# Multimodal profiling reveals tissue-directed signatures of human immune cells altered with age

Steven B. Wells [1,2,13], Daniel B. Rainbow [3,13], Michal Mark [4,13],
Peter A. Szabo [2,13], Can Ergen [5,13], Daniel P. Caron [2,13],
Ana Raquel Maceiras [6,7,13], Elior Rahmani[5], Eli Benuck[4],
Valeh Valiollah Pour Amiri[5], David Chen [1], Allon Wagner [5],
Sarah K. Howlett[3], Lorna B. Jarvis [3], Karen L. Ellis[3], Masaru Kubota[8],
Rei Matsumoto[8], Krishnaa Mahbubani [9], Kouresh Saeb-Parsy [9],
Cecilia Dominguez Conde[6], Laura Richardson [6], Chuan Xu [6,7], Shuang Li[6,7],
Lira Mamanova[6], Liam Bolt[6], Alicja Wilk[6,7], Sarah A. Teichmann [7,10,11,14 ✉],
Donna L. Farber [2,8,14 ✉], Peter A. Sims [1,12,14 ✉], Joanne L. Jones [3,14 ✉] &
Nir Yosef [4,5,14 ✉]

The immune system comprises multiple cell lineages and subsets maintained in tissues throughout the lifespan, with unknown effects of tissue and age on immune cell function. Here we comprehensively profiled RNA and surface protein expression of over 1.25 million immune cells from blood and lymphoid and mucosal tissues from 24 organ donors aged 20–75 years. We annotated major lineages (T cells, B cells, innate lymphoid cells and myeloid cells) and corresponding subsets using a multimodal classifier and probabilistic modeling for comparison across tissue sites and age. We identified dominant site-specific effects on immune cell composition and function across lineages; age-associated effects were manifested by site and lineage for macrophages in mucosal sites, B cells in lymphoid organs, and circulating T cells and natural killer cells across blood and tissues. Our results reveal tissue-specific signatures of immune homeostasis throughout the body, from which to define immune pathologies across the human lifespan.

The immune system leverages a dynamic network of specialized cells spread across the body to defend against infections and cancer, regulate inflammation and repair tissue damage. Myeloid cells—macrophages, monocytes and dendritic cells (DCs)—initiate innate immunity at mucosal and barrier sites, while adaptive immunity is mediated by antigen-specific T and B lymphocytes in lymphoid organs. Immune memory is established following antigen-driven activation and differentiation of T cells and B cells, resulting in heterogeneous subsets of circulating and tissue resident memory T cells ($T_{RM}$) and B cells that

persist in diverse tissues[1]. With age, immune memory accumulates, although responses can become dysregulated, increasing susceptibility to infections, cancer and autoimmunity. As human immune cells and the effect of age are mostly studied in blood[2], we lack a comprehensive understanding of the effect of age on the majority of innate and adaptive immune cells that are maintained in tissues.

Investigating human tissue immunity across diverse ages has been difficult to achieve. Obtaining tissues from organ donors enables the acquisition of blood and multiple tissues from individual

A full list of affiliations appears at the end of the paper. ✉e-mail: sat1003@cam.ac.uk; df2396@cumc.columbia.edu; pas2182@cumc.columbia.edu; jls53@medschl.cam.ac.uk; nir.yosef@weizmann.ac.il

donors and the isolation of viable immune cells across lineages for phenotypic, functional and multimodal single-cell profiling. Previous studies of lymphocytes from organ donors showed that T lymphocyte, natural killer (NK) cell and innate lymphoid cell (ILC) subset composition, tissue residence and certain functional attributes are specific to the tissue[3-9], indicating that localization has a dominant role in determining the maintenance and functional responses of lymphocytes. Whether these tissue-specific effects on human T cells are exhibited by other immune cell lineages, such as B cells and myeloid cells, and whether aging exerts general or tissue-specific effects have not been established.

Here, we present a comprehensive analysis of human immune cells using cellular indexing of transcriptomes and epitopes (CITE-seq) to simultaneously profile transcriptomes and >125 surface proteins of myeloid and lymphoid-lineage cells in 14 tissue sites of 24 organ donors aged 20–75 years. We identified a crucial role of tissue on immune cell composition, function, homing and differentiation across myeloid and lymphocyte lineages, including signatures specific for the gut, lungs and lymphoid organs. Across age, site-specific immune cell composition was largely maintained, although age-associated changes in function, signaling and metabolism were identified in certain subsets and sites, including macrophages in the lung, B cells in lymphoid organs and CD8+ T cells across sites. Together, our findings reveal the complex interplay between tissue, lineage, subset and age in immune homeostasis that is important for defining immune dysfunctions in disease.

## Results

### Multimodal profiling identifies immune cells across tissues

We isolated mononuclear cells (MNCs) from blood, multiple lymphoid organs, lungs, airways, intestines and other sites using established protocols[7,9] from 24 donors (10 females and 14 males, aged 20–75 years) (Fig. 1a). Organ donors originated from New York City (USA) and Cambridge (UK) and were free of chronic infection, cancer and overt disease (Supplementary Table 1). We performed single-cell RNA sequencing (scRNA-seq) on tissues from all donors, including CITE-seq with 127 proteins from 22 donors (Supplementary Table 2). We obtained 1.28 million immune cell events from 10 sites with >75,000 cells per site, including blood, bone marrow (BM), spleen, different lymph nodes (LNs) including lung-associated LN (LLN), mesenteric LN (MLN) and inguinal LN (ILN), lungs, comprising bronchoalveolar lavage (BAL) and lung parenchyma, and jejunum (JEJ), divided into the intraepithelial layer (JEL) and lamina propria (JLP) (Fig. 1a). We also purified low numbers of immune cells from liver, skin, colon epithelium and colon lamina propria from 9 donors (4 females, aged 25–75 years; 5 males, aged 20–55 years). These sites were therefore not included in the annotated dataset below and are provided as a separate reference (Extended Data Fig. 1a).

For data integration, we leveraged multi-resolution variational inference (MrVI), which is designed for cohort studies. MrVI harmonizes variation between cell states (for unified annotation of cell states across samples) and accounts for differences between samples[10]. Visualization with uniform manifold approximation and projection (UMAP)[11] showed similar results across US and UK donors, sequencing technologies and other donor covariates such as sex, cytomegalovirus (CMV) and Epstein–Barr virus serostatus (Fig. 1b and Extended Data Fig. 1b–g). Although cells from blood and lymphoid organs (BM, spleen, LNs) clustered similarly, cells from mucosal sites (lung, JEJ) clustered distinctly (Fig. 1c). For annotation, we used MultiModal Classifier Hierarchy (MMoCHi)[12], which leverages both surface protein and gene expression to hierarchically classify cells into predefined categories (Supplementary Fig. 1 and Supplementary Table 3). MMoCHi defined six major immune lineages found across all tissues (Fig. 1d) comprising 13 T cell, 5 NK/ILC, 6 B cell and 7 myeloid subsets (Extended Data Fig. 1h). These results showed broad and consistent representation of major immune lineages in our dataset.

### Immune cell subset composition is specific to the tissue

We next analyzed the subset composition and heterogeneity for each immune cell lineage across tissues based on the MMoCHi annotations above. T lymphocytes (610,429 cells) comprised low-frequency γδ T cells, which develop early in ontogeny, and predominant αβ T cells (Fig. 2a,b and Supplementary Fig. 2). CD4+ and CD8+ T cells (αβ TCR+) were subdivided into naive ($T_N$), terminal effector ($T_{EMRA}$) and memory subsets, including effector-memory ($T_{EM}$), central memory ($T_{CM}$) and $T_{RM}$, along with CD4+ regulatory T cells ($T_{reg}$)[6,13,14] (Fig. 2a,b). Surface proteins were essential for identifying T cell subsets that were not fully resolved by scRNA-seq (Fig. 2b and Supplementary Fig. 3), as shown before[9,12,15]. For example, surface CD45RA expression was required to distinguish CD45RA+ $T_N$ cells from CD45RA− $T_{CM}$ cells and CD45RA− $T_{EM}$ cells from CD45RA+ $T_{EMRA}$ cells, and surface γδ or αβ T cell receptor (TCR) expression to accurately identify γδ T cells from CD8+ T cells, which can express *TRDC*[12] (Fig. 2b). In addition, $T_{RM}$ cells were distinguished from $T_{EM}$ cells based on surface expression of CD69, CD103 and/or CD49a[6] (Fig. 2b and Supplementary Fig. 3). T cell subsets were differentially distributed across sites; CD4+ $T_N$, CD8+ $T_N$ and CD4+ $T_{CM}$ cells were enriched in blood and multiple LN, CD4+ $T_{reg}$ cells were enriched in LN, while CD4+ and CD8+ $T_{RM}$ cells predominated in JEJ and were present at lower frequencies in lungs, spleen and LN (Fig. 2a,b). $T_{EMRA}$ cells were mostly CD8+ and enriched in the BM and spleen and, to a lesser extent, in lungs, while $T_{EM}$ cells were distributed across most sites (Fig. 2a,b). Mucosal-associated invariant T (MAIT) cells, distinguished by *TRAV1.2* expression, CD161 and other markers[16], were predominantly found in the spleen, BM and lungs (Fig. 2a,b). TCR clonal analysis provided additional correlative evidence for subset delineation and tissue distribution (for example, with the highest clonality observed in the $T_{EMRA}$ subset, as previously described[7,17]) (Supplementary Fig. 2).

Innate lymphocytes (130,414 cells) were predominantly mature CD56dimCD16+ NK cells expressing cytolytic markers (*KLRF1*, *GZMB*) and enriched in blood, BM and lungs (Fig. 2c,d). Immature CD56brightCD16− NK cells were present at lower frequencies across most tissues (Fig. 2c,d). We detected low frequencies of ILCs consisting largely of CD16−NCR2+IL7R− ILC1s with high expression of tissue residency markers (CD69, CD49a, CD103) enriched in JEJ, and IL7R+KIT+RORC+ ILC3 (ref. 8) found in LN, spleen and JEJ (Fig. 2c,d). Putative CD127+CD62L+TCF7+ NK/ILC precursors, resembling CD56brightCD16− NK and ILC3s[18], were enriched in blood but present in all tissues (Fig. 2c,d).

B cells (272,162 cells) were classified into 6 subsets largely confined to lymphoid organs (Fig. 2e,f), including IgD+ naive B cells, CD27+ memory B cells, germinal center B cells expressing *AICDA* (encoding activation-induced adenosine deaminase, which mediates somatic mutation and class switch recombination[19]), plasma cells expressing immunoglobulin genes and *SDC1* (CD138) and plasmablasts expressing proliferation markers (*MKI67*, *TOP2A*) (Fig. 2f). CD11c+ memory B cells expressing *TBX21* (encoding the transcription factor T-BET) resembled 'atypical B cells'[20] and were found at low frequencies in spleen and BM (Fig. 2e,f). Variable frequencies of memory B cells in LN and spleen expressed CD69 (Fig. 2e,f), denoting tissue residency[21]. Plasma cells expressing IgA were enriched in the JLP, while IgG+ plasmablasts were enriched in lymphoid organs (Fig. 2e,f). B cell receptor (BCR) analysis indicated that plasmablasts exhibited the highest clonal expansion across lymphoid sites, while memory B cells and plasma (but not naive) cells expressed mutated BCR (Supplementary Fig. 4).

Myeloid lineage cells (225,268 cells) comprised *C1QA+MS4A7+* macrophages, *FCN1+CD14+* classical and *FCN1+FCGR3A+* non-classical monocytes and DC subsets, including *CLEC9A+* DC1, *CLEC10A+* DC2, *CCR7+* migratory DCs and CD123+*LILRA4+* plasmacytoid DCs (pDCs) (Fig. 2g,h). Classical monocytes were most abundant in blood, while non-classical monocytes were found mainly in BM and lung (Fig. 2g,h). DCs were found at low frequency across all tissues; pDCs were enriched in the BM, migratory DCs in the LNs, while DC2s were found mainly in

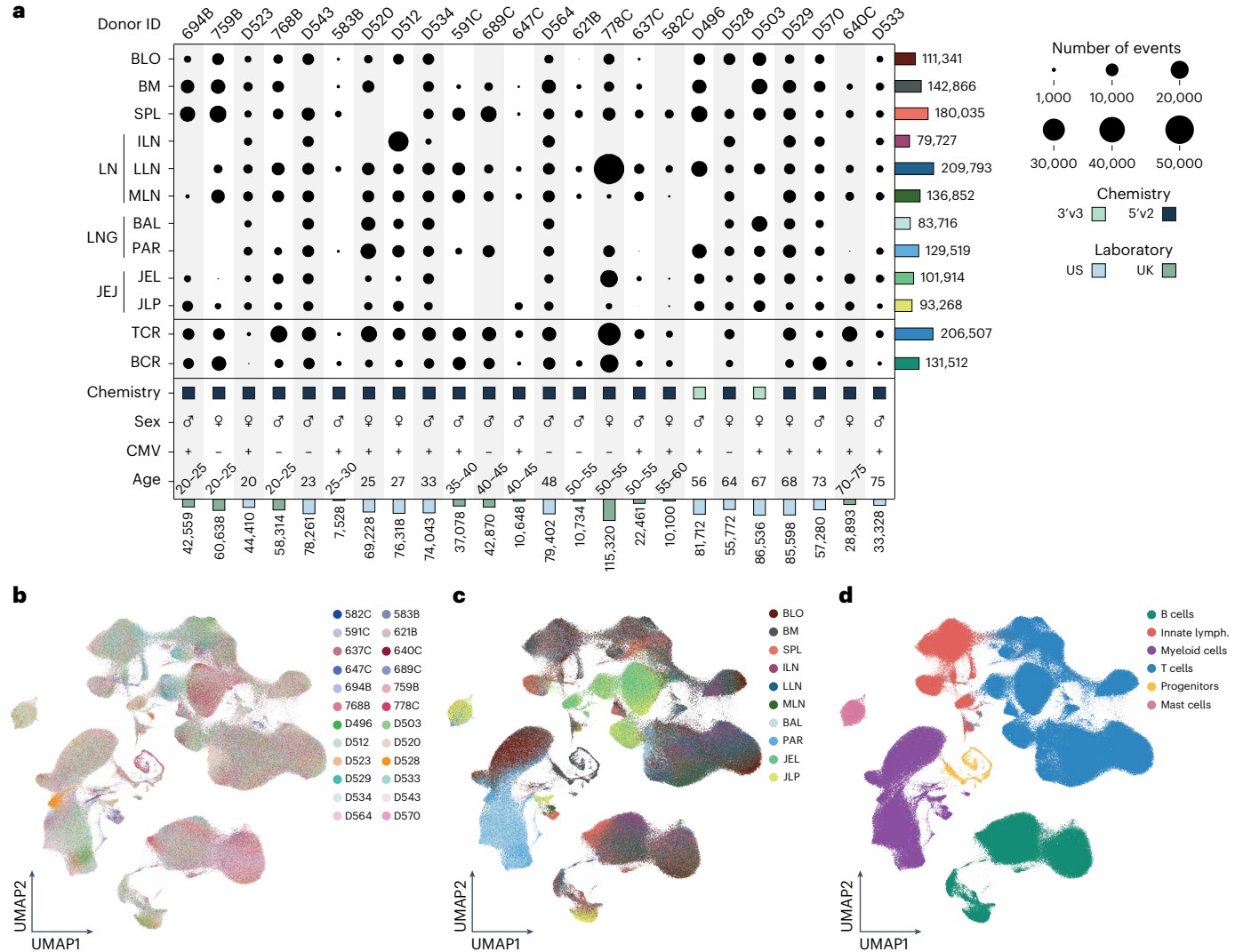

**Fig. 1 | A multi-lineage human immune cell atlas encompasses tissues and age. a**, Plot of cell numbers across 24 donors and 10 tissue sites (blood (BLO); BM; spleen (SPL); LNs, including ILN, LLN and MLN; lung (LNG), comprising BAL and parenchyma (PAR); and JEJ (divided into JEL and JLP)) and donor metadata including 10× Genomics sequencing chemistry (3′, n = 2; 5′, n = 22), sex (male, n = 14; female, n = 10), CMV serostatus (positive, n = 16; negative, n = 8), age

(range, 20–75 years) and location of tissue acquisition (US, n = 12; UK, n = 12). Bottom bars depict the number of cells profiled from each donor, and right bars depict the number of cells in each tissue across all donors. **b–d**, UMAP embeddings colored by donor (**b**), tissue site (**c**) or immune lineage classified by MMoCHi (**d**) in the BLO, BM, SPL, ILN, LLN, MLN, BAL, PAR, JEL and JLP. Lymph., lymphocytes.

the LNs and JEJ (Fig. 2g,h). Macrophages were found predominantly in the lung and at low frequencies in the BM, spleen, LNs and JEJ (Fig. 2h).

Immune cell subset composition for the lineages above in blood, lymphoid organs and mucosal sites was specific to the site and conserved across donors (Fig. 2i and Extended Data Fig. 2). We defined additional subset heterogeneity based on RNA expression, identifying proliferating cells across lineages and functional subsets for CD8⁺ $T_{EM}$ cells and CD8⁺ $T_{EMRA}$ cells, also with tissue-specific distribution (Supplementary Figs. 5 and 6). This comprehensive, annotated map of immune cells across tissues can serve as a reference for future analysis, and we provided pre-trained models for label transfer of our cell-type annotation in the popV framework[22] (Supplementary Fig. 7). This subset analysis showed consistent, tissue-specific composition between sites.

**Immune cell phenotype is strongly tissue-associated**

To understand the influence of tissue localization on gene expression, we performed a two-step differential expression (DE) analysis. First, we compared the major immune lineages (CD8⁺ T cells, CD4⁺ T cells, γδ/MAIT cells, myeloid cells, NK/ILC and B cells) within each site (for

example, blood, BM, spleen, LN, lung, JEJ) versus the other sites using pseudobulked linear mixed modeling (LMM), controlling for covariates (for example, sex, CMV serostatus)[23] (Fig. 3a and Supplementary Tables 4 and 5). We identified 13 clusters from significantly differentially expressed genes (DEGs) (C1–13) grouped by lineage and/or tissue (Fig. 3b and Supplementary Table 6). We then conducted a similar across-tissue DE analysis for each subset and evaluated whether these signatures were expressed by individual subsets within a lineage using gene set enrichment analysis (GSEA) (Fig. 3a). To visualize effect sizes, determine the contribution of compositional differences and identify tissue-specific signatures across lineages, we integrated this analysis with subset frequencies within a tissue, fold changes (FC) compared to the other sites and the average expression of gene clusters by subset (Fig. 3c–h, Extended Data Fig. 3, and Supplementary Tables 7 and 8).

Myeloid cells from all tissues exhibited transcriptional profiles not expressed by other immune lineages (clusters C1–C4), which also varied by tissue (Fig. 3b). C1 (chemokine, complement, lipid transport) and C2 (PPAR signaling associated with alveolar macrophages[24]) were enriched in the JEJ and lung, respectively, particularly in

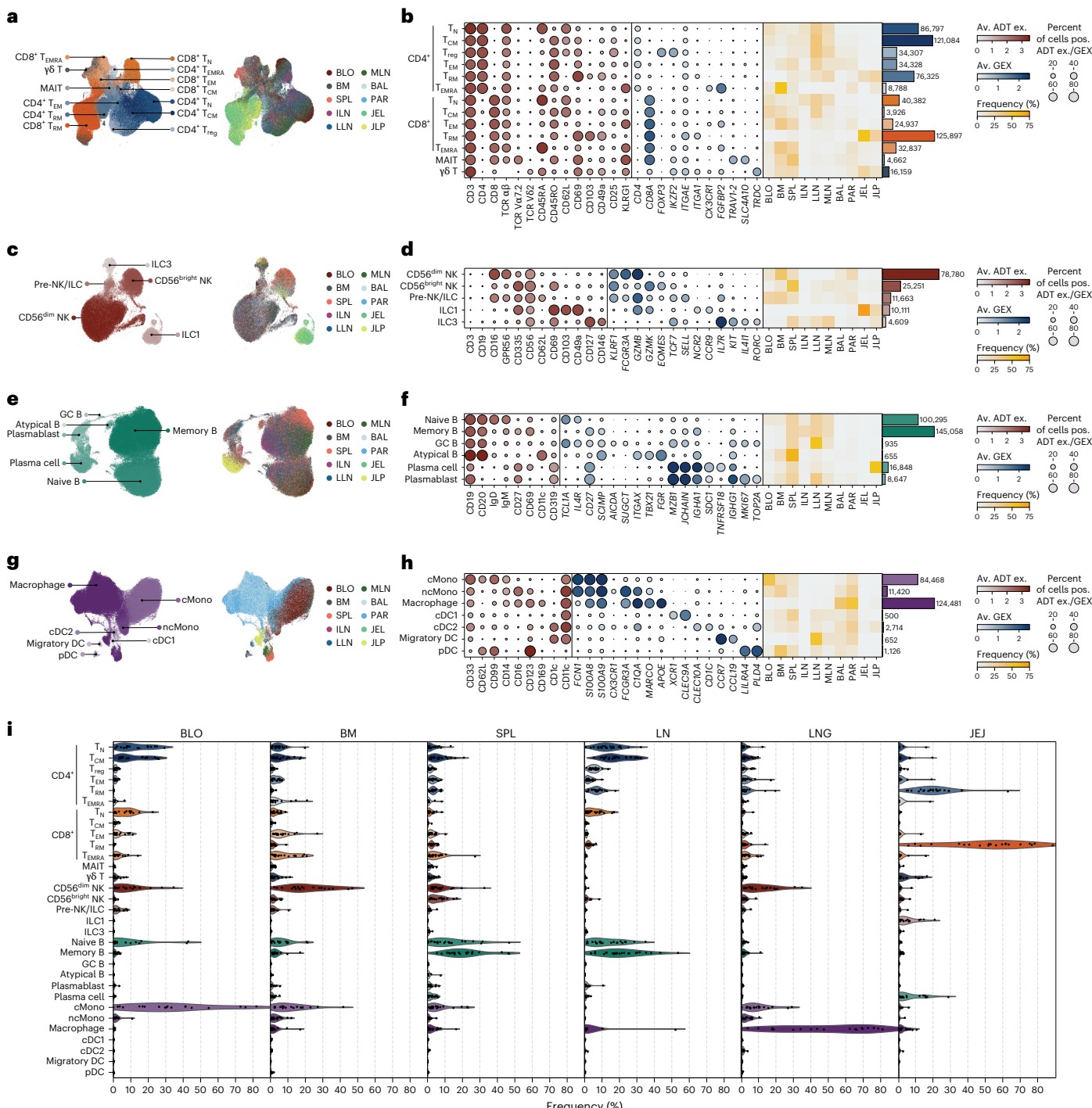

**Fig. 2 | Multimodal classification reveals immune subset heterogeneity and distribution across sites. a–h**, MMoCHi classification of immune cell subsets among T cells (**a** and **b**), NK/ILCs (**c** and **d**), B cells (**e** and **f**) and myeloid cells (**g** and **h**) in the BLO, BM, SPL, LNs (ILN, LLN, MLN), lung (BAL, PAR) and JEJ (JEL and JLP) shown as UMAP embeddings (**a**, **c**, **e** and **g**), with immune cell profiles colored by MMoCHi subset classification (left) or tissue of origin (right), or as heatmaps (**b**, **d**, **f** and **h**) showing percentage positive surface protein expression (red dot plot), averaged gene expression (blue dot plot) and relative

frequency distribution (rows sum to 100%) across tissues (yellow heatmap) for subsets. Bars on the right depict cell numbers for each subset. **i**, Violin plots of immune subset composition in BLO, BM, SPL, LNs (including ILN, LLN and MLN), lung (including PAR and BAL) and JEJ (including JEL and JLP). Dots represent the frequency of a subset within each donor (frequencies sum to 100% for each donor). Av., average; ex., expression; pos., positive; GC B, germinal center B cell; cMono, classical monocyte; ncMono, non-classical monocyte; cDC, classical dendritic cell.

macrophages (Fig. 3c,d). By contrast, C4 (anti-microbial peptide production and cell signaling) enrichment was explained by increased frequencies of BM and spleen monocytes relative to macrophages rather than transcriptional changes within and across subsets (Extended Data Fig. 3a,b).

Tissue-associated genes within clusters C5–C9 were expressed primarily by T cells and innate lymphocytes (Fig. 3b). C5 genes encoding stem-like transcription factors and markers (*TCF7*, *LEF1*, *ITGA6*)[25,26] and LN homing receptors (*CCR7*, *SELL*) were enriched in CD4+ and CD8+ TRM cells and TEM cells in LN (Fig. 3e,f). Conversely,

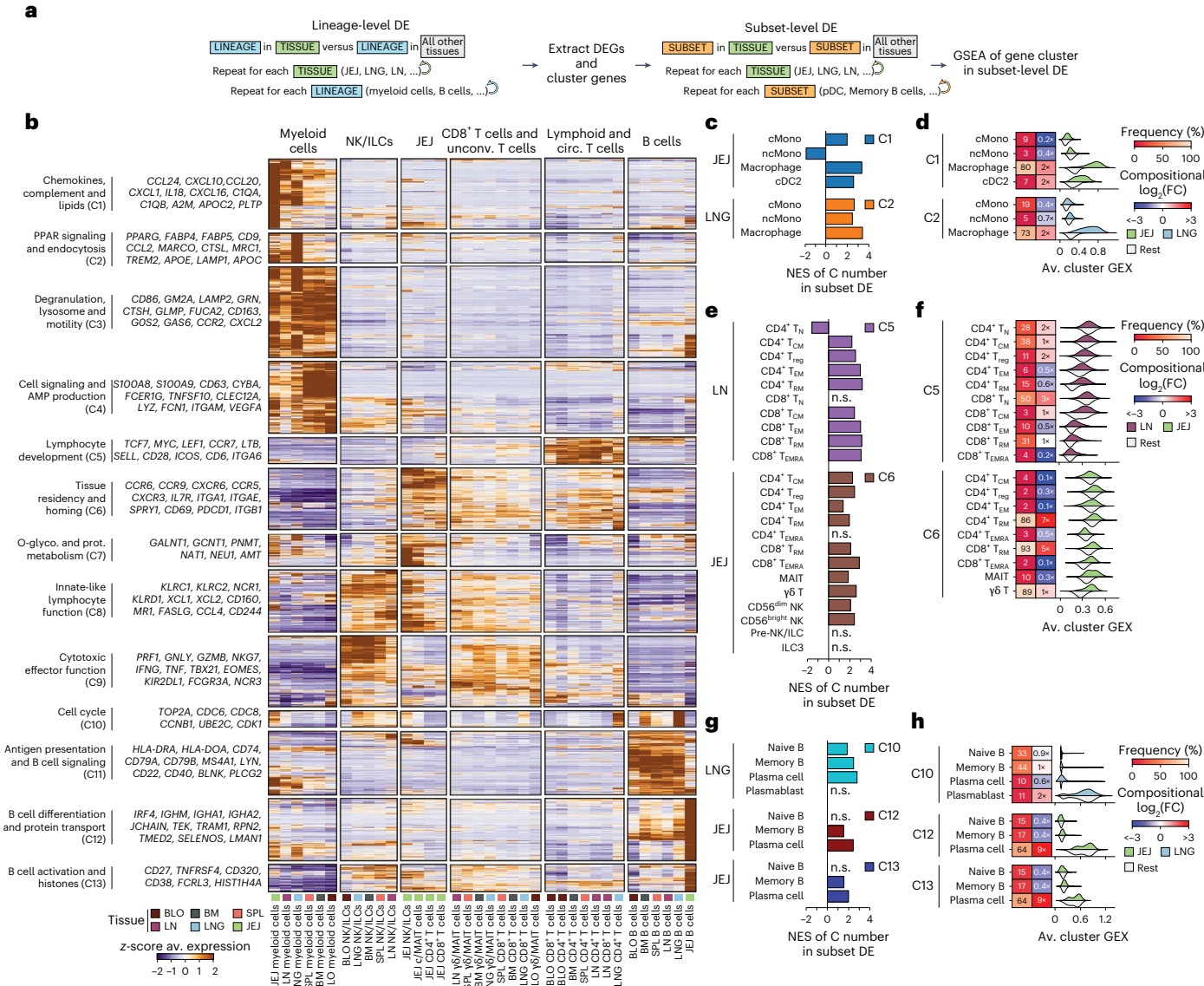

**Fig. 3 | Differential expression analysis identifies tissue localization as a major determinant of immune cell identity across lineages. a,** Schematic showing cell lineage-level and cell subset-level DE analysis using linear mixed models (evaluated by dreamlet) in which cell lineage-level analysis compares a lineage in one tissue (for example, B cells from JEJ) against the same lineage in all other tissues, and cell subset-level analysis compares an immune cell subset in one tissue (for example, plasma cells in JEJ) against the same immune cell subset in all other tissues, followed by pre-ranked GSEA to assess gene cluster enrichment in subsets across tissues. **b,** Heatmap of lineage-level DEGs (adjusted (adj.) $P$ value (false discovery rate, FDR) < 0.05, $\log_2$(FC) > 1, average mean expression >2) showing $z$-score average gene expression across

tissue–cell lineage combinations, labeled by gene cluster with selected genes. **c–h,** Evaluation of gene clusters in cell subset-level DE analysis for myeloid cells (**c** and **d**), T cells and NK/ILCs (**e** and **f**), and B cells (**g** and **h**), shown as bar plots (**c, e** and **g**) with normalized enrichment scores (NES) of indicated gene clusters in cell subset-level DE by pre-ranked GSEA and heatmaps (**d, f** and **h**) showing cell subset frequencies as a percentage of their lineage within each tissue, FC in subset frequency in indicated tissue versus other tissues and split violin plots showing average gene cluster expression in the indicated tissue versus other tissues. Plotted bars denote adj. $P$ value (FDR) < 0.05; n.s., not significant. AMP, adenosine monophosphate; O-glyco., O-glycosylation; prot., protein; circ., circulating; unconv., unconventional.

C6 genes encoding molecules for mucosal residency (*ITGA1*, *CXCR6*, *ITGAE*)[6] and gut homing (*CCR9*) were enriched in CD4$^+$ and CD8$^+$ T$_{RM}$ cells, γδ T cells, MAIT cells and NK cells in the JEJ (Fig. 3e,f). CD8$^+$ T$_{EMRA}$ cells, γδ cells, MAIT cells and NK/ILCs in the JEJ showed reduced expression of C9 genes associated with cytolytic effector function (*GZMB*, *PRF1*, *IFNG*, *NKG*7) compared to other sites, while C9 was enriched in the lung (Fig. 3b and Extended Data Fig. 3c,d). C7 genes associated with protein metabolism were enriched in NK/ILCs and innate T cells in the JEJ, while C8 genes encoding chemokines (*CCL4*, *XCL1*) and innate lymphocyte functions (*KLRC1*, *NCR1*) were expressed by T cell and NK/ILC subsets across all sites (Fig. 3b and

Extended Data Fig. 3e,f). Therefore, this analysis identified shared gene expression profiles across T and innate lymphocyte subsets that varied by site.

Tissue-enriched signatures for B cells (C10–C13) were mostly associated with specific subsets (Fig. 3b,g,h and Extended Data Fig. 3g,h). C10 included cell cycle genes and was enriched in B cells and plasma cells in the lungs (Fig. 3g,h), possibly because of higher plasmablast abundance. JEJ B cells showed increased expression of C12 (B cell differentiation and protein transport) and C13 (B cell activation) genes (Fig. 3b), which were derived from plasma cells comprising the majority of the B cell lineage in intestinal sites (Fig. 3g,h).

We also applied consensus single-cell hierarchical Poisson factorization (scHPF)[15,27] to identify gene co-expression patterns common to different immune lineages or sites (considering the JEJ, lung and LN) (Extended Data Fig. 4a and Supplementary Table 9). We found a proliferation module enriched in the lungs, a lymphoid-specific module in the LNs, an intestine-specific residency module and modules associated with effector and cytolytic functions prominent in the lungs but not JEJ or LNs (Extended Data Fig. 4a–d). The lymphoid module included genes associated with stemness (*KLF7*, *LEF1*, *SOX4*)[28,29] and lymphoid homing (*CCR7*, *SELL*, *ITGA6*) markers expressed mostly in T cells, some NK cells and ILCs, and not in B cells or myeloid cells (Extended Data Fig. 4b–e and Supplementary Table 10). The JEJ residency module included intestinal tissue residency genes (*CCR9*, *ITGAE*, *CD101*, *CD160*)[7,30] enriched in T cells, NK cells and ILCs (Extended Data Fig. 4f–h). These signatures were also reflected at the surface protein level (Extended Data Fig. 4e,i and Supplementary Table 11). Thus, tissue-specific gene expression modules spanned multiple cell types, suggesting shared tissue adaptations.

## Resident immune cells express tissue-specific signatures

The tissue environment poses unique requirements for resident immune populations such as $T_{RM}$ cells, plasma cells and macrophages for maintaining homeostasis[31–33]. We integrated the DE analysis shown in Fig. 3 with surface protein expression to define site-specific signatures for resident immune cells (Fig. 4 and Supplementary Table 12). CD4[+] and CD8[+] $T_{RM}$ cells expressed genes and/or surface proteins for tissue residency (CD103, CD101, CD49a), gut homing and localization (*CCR9*, CCR5 for CD4[+] $T_{RM}$ cells) and reduced PD-1 (for CD8[+] $T_{RM}$ cells) in intestines relative to lungs and lymphoid organs (Fig. 4a–f). Lung $T_{RM}$ cells showed increased expression of effector or cytotoxicity (*IFNG*, *GZMH*, *GZMA*, *PRF1*) and regulatory (*CTLA4*) genes relative to the JEJ and lymphoid organs; $T_{RM}$ cells in LN, spleen and BM had higher expression of stem-like markers (*TCF7*, *KLF2*) and certain integrins and costimulatory markers (*ITGB2*, CD27, CD28, ICOS) relative to the JEJ and lungs (Fig. 4a–f). These site-specific profiles for $T_{RM}$ cells showed adaptations related to migration, localization and function.

Tissue plasma cells and macrophages also exhibited distinct tissue signatures. For plasma cells, we found several JEJ-enriched genes, including *IGHA2* (consistent with predominant IgA[+] plasma cells in the gut), the non-classical HLA molecule CD1d, plasma cell transcription factors (*RUNX2*, *ID3*)[34,35] and the tissue residency marker CD69 (Fig. 4g–i). Plasma cells in lungs, and to a lesser extent, LN and spleen, expressed higher levels of integrins (CD11c, CD18) (Fig. 4g–i). JEJ macrophages had increased expression of integrins (*ICAM1*, *ITGA4*), chemokines (*CCL24*, *CCL3L1*) and regulators of macrophage activation (for example, *SLAMF7*)[36], whereas lung macrophages had higher expression of CD11c (*ITGAX*), markers of efferocytosis (*MRC1*, *MARCO*) and lipid metabolism (*PPARG*, *TREM2*, *FABP4*) (Fig. 4j–l), consistent with alveolar macrophages[37]. Lastly, macrophages in the spleen expressed markers of red pulp macrophages (*SPIC*, *VCAM1*)[38] (Fig. 4j). These observations revealed site-specific signatures for activation, migration, metabolism and cell–cell interactions involved in tissue residency.

## Immune cell profiles change with age across tissues

We investigated age-associated effects across immune lineages and tissues. A global analysis of transcriptional variance within each major lineage revealed that tissue site explained the majority of variance, while age accounted for a much smaller fraction (Fig. 5a and Supplementary Table 13). Most of the top age-dependent genes in each lineage (Fig. 5b) also exhibited tissue-specific variation. Immune cell composition in the different sites was largely maintained across age, except for significantly decreased frequencies of CD8[+] $T_N$ cells in the blood and LNs, concomitant increases in $T_{EM}$ cells in the blood and $T_{RM}$ cells in the LNs and lower frequencies of classical monocytes in the BM (Fig. 5c

and Supplementary Table 14). These results indicate that tissue-driven immune cell composition and profile are largely maintained with age.

To interrogate specific effects of aging on immune cells across lineages and sites, we conducted a separate DE analysis for each tissue group and immune subset with sufficient representation, comparing younger (<40 years) versus older (>40 years) individuals, while controlling for sex, CMV status and other covariates (Supplementary Tables 15–17). An embedding of similarities between age-related DEGs revealed that some immune cells exhibited changes specific to tissue site (for example, T cells in the LN, lymphocytes in the JEJ), while others showed subset-specific changes independent of site (for example, monocytes in blood, BM and lung, and memory B cells in LNs and lung) (Fig. 5d).

We identified genes regulated with age in each subset and tissue by integrating DE results with GSEA (Supplementary Fig. 8a). For myeloid lineage cells, mucosal sites had the most age-related DEGs, with similar trends in lymphoid organs (Fig. 5e–g). Age-related changes in classical monocytes included increased expression of genes associated with proliferation and inflammation (*KRAS*, *CALM1*)[39,40] and decreased expression of genes for macrophage differentiation (for example, *RAB44*)[41] (Fig. 5e); non-classical monocytes showed age-associated increase in expression of genes for cell–cell interactions (*LGALS1*, *ITGA9*) and decreased expression of metabolism and mitochondrial regulation genes (*LYRM7*, *SARS2*) (Fig. 5f). Macrophages had decreased expression of genes associated with metabolism and mitochondrial fitness (*MARC1*, *MTOR*)[42] and increased expression of genes associated with M2 macrophages (*ID1*, *ADGRE1 NPR2*)[43,44] and interferon signaling (*MX2*) (Fig. 5g) along with increased CD95 (Fas) at the protein level (Fig. 5h). Together, these age-related changes in monocytes and macrophages were subset-specific, enriched in mucosal sites and indicated decreased overall fitness.

We used scHPF to identify age-associated gene signatures for myeloid cells and GSEA to assess their expression in different subsets and sites (Supplementary Tables 18 and 19). We uncovered an *APOE–TREM2* signature, including apolipoprotein genes (*APOC1*, *APOC2* and *APOE*) and *TREM2*, a triggering receptor expressed on myeloid cells[45] that binds ApoE and facilitates macrophage functions, such as phagocytosis and chemotaxis, and induces metabolic changes[46] (Fig. 5i). This *APOE–TREM2* signature was significantly downregulated with age in monocytes and macrophages in the lungs, lymphoid organs and blood (Fig. 5j), and in an independent published dataset from human lungs[47] (*n* = 29; see Methods) (Fig. 5k). Overall, this analysis showed subset and site-specific features of macrophage aging involving a major functional and metabolic pathway.

## T and B lymphocytes express distinct aging signatures

We applied the above approaches to identify age-associated signatures in T cells and B cells. For CD4[+] T cells, CD8[+] T cells and B cells, there were relatively few genes (for example, those associated with oxidation and inflammation: *SOD1*, *IL18BP*, *IL15*) that changed over age in two or more subsets within each lineage (Extended Data Fig. 5a–d). Pathway analysis revealed increased inflammation, apoptosis and reduced TCR signaling across multiple T cell subsets with age (Supplementary Fig. 8b). Despite the paucity of age-associated gene expression changes across subsets, CD8[+] $T_{EMRA}$ cells exhibited multiple age-associated changes conserved across sites; these included increased expression of NK cell genes (*NCAM1*, *KLRF1*, *GNLY*) and the NK cell marker CD56, consistent with findings in blood[48], and reduced expression of genes associated with signaling (*CD6*, *JAK3*), proliferation and metabolism (*TCF7*, *RPTOR*) (Fig. 6a,b).

By scHPF analysis, we identified two prominent age-associated transcriptional signatures shared across multiple CD8[+] T cell subsets. A cytokine signature, containing genes for effector cytokines and chemokines (for example, *CCL3*, *CCL4*, *XCL1*, *IFNG*, *TNF*), was increased with age in all CD8[+] T cell subsets and γδ T cells across all sites examined

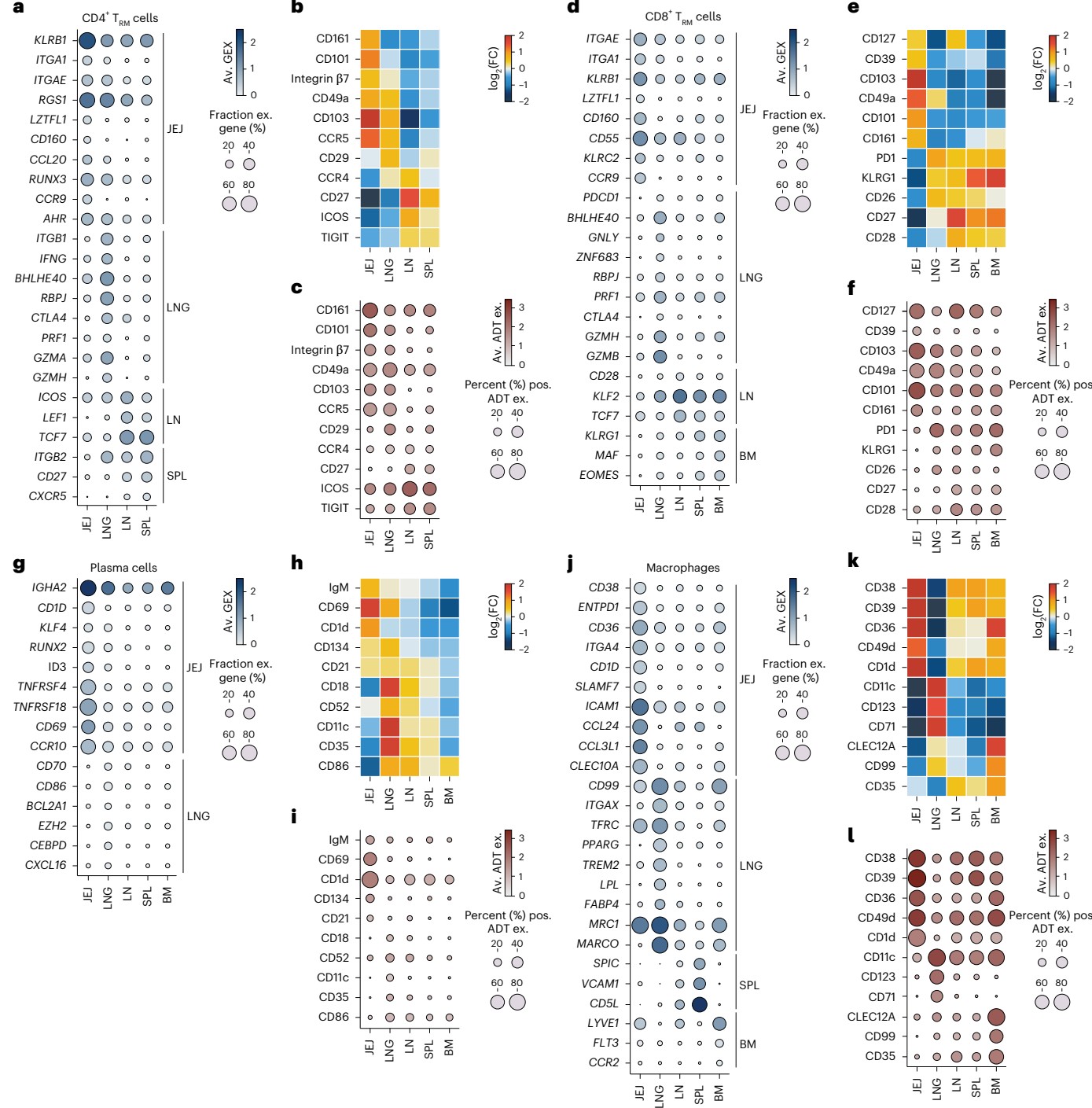

**Fig. 4 | Cross-tissue analysis of resident immune cell subsets highlights site-specific signatures. a–l,** Expression of tissue-enriched genes from cell subset-level DE analysis and corresponding surface proteins for CD4⁺ T$_{RM}$ cells (**a–c**), CD8⁺ T$_{RM}$ cells (**d–f**), plasma cells (**g–i**) and macrophages (**j–l**), shown as dot plots with the percentage of cells expressing selected genes enriched in one tissue versus the other sites (**a, d, g** and **j**) or surface protein expression (**c, f, i** and **l**) and heatmaps showing median log$_2$(FC) surface marker expression across tissues (**b, e, h** and **k**). Dot size represents the frequency of gene expression in group (**a, d, g** and **j**) or frequency (% positive) (**c, f, i** and **l**); color (red, blue) intensity indicates average gene expression.

(Fig. 6c,d). This aging signature was also detected in published datasets from human lungs[47] (Fig. 6e) and in peripheral blood mononuclear cells (PBMCs) from the Sound Life cohort (*n* = 96, age 26–65 years) within the Human Immune Health Atlas[49] (Extended Data Fig. 5e). A second signature contained *GZMK* encoding the cytolytic molecule granzyme K, the transcription factor *EOMES*[50] and activation or signaling markers (*PDCD1*, *HLA-DR*, *FCRL3*)[51] (Fig. 6f). This signature aligned with a granzyme K-containing, age-associated signature identified in

T cells in aged mice and human blood[52] and in PBMCs from the Human Immune Health Atlas[49] (Extended Data Fig. 5f). This *GZMK*⁺ signature was increased with age in CD8⁺ T$_{EMRA}$ cells, γδ T cells and CD8⁺ T$_{EM}$ cells across tissues but not in CD8⁺ T$_{RM}$ cells in lungs and JEJ (Fig. 6g). The frequency and clonal expansion of CD8⁺ T$_{EMRA}$ cells enriched in the *GZMK*⁺ signature was higher in older than in younger donors (Fig. 6h,i). These results show distinct age-associated signatures in mucosal resident T cells compared to circulation.

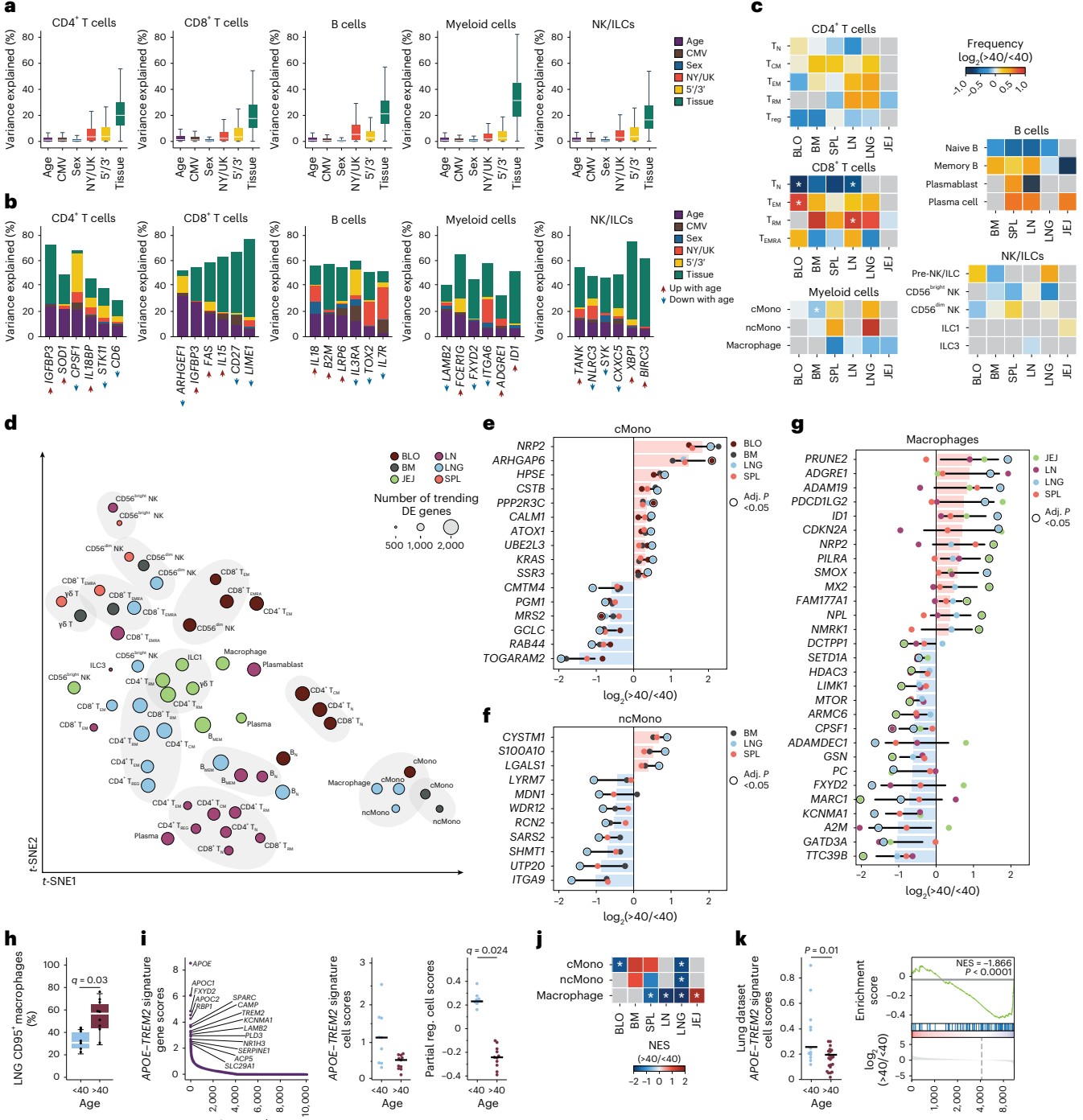

**Fig. 5 | Immune aging manifests across lineage, subset and tissue of origin.**
**a**, Variance decomposition analysis (dreamlet), with box plots showing percentage of variance in gene expression explained by age (<40 or >40 years old), CMV serostatus (positive or negative), sex (male or female), processing site (US or UK), 10× Genomics chemistry (5′ or 3′) and tissue group (BLO, BM, SPL, LN, LNG or JEJ) for CD4⁺ T cells, CD8⁺ T cells, B cells, myeloid cells and NK/ILCs. Box plots show the median (center), interquartile range (IQR; box) and whiskers extending to 1.5× IQR. **b**, Variance decomposition analysis with stacked bar plots showing selected genes from the top (up with age) and bottom (down with age) 50 DEGs by cell lineage-level linear mixed model (dreamlet) across age for CD4⁺ T cells, CD8⁺ T cells, B cells, myeloid cells and NK/ILCs. **c**, Heatmaps of change in subset composition by age (<40 or >40 years old) across BLO, BM, SPL, LN, LNG and JEJ, shown as log₂(FC) calculated by generalized linear model (two-sided Wald test). Gray denotes insufficient cell numbers for comparison. **d**, t-SNE of immune cell subsets based on age-associated DEGs (log₂(FC) > 0.1, unadjusted (unadj.) P < 0.05) by subset-level linear mixed model (dreamlet).

**e–g**, Bar-and-dot plots of top DEGs by age (adj. P value (FDR) < 0.05 for at least 1 tissue; among top 50) for classical monocytes in BLO, BM, LNG and SPL (**e**), non-classical monocytes in BM, LNG and SPL (**f**), and macrophages in JEJ, LN, LNG and SPL (**g**). Solid bars show median log₂(FC) across tissues; error bars, 95% CI. Statistically significant genes are indicated by circles outlined in black. **h**, Box plot showing the percentage of CD95⁺ macrophages in the lung in donors <40 or >40 years old. Statistical significance determined by generalized linear model (two-sided Wald test). **i**, Consensus scHPF macrophage APOE–TREM2 signature with dot plots showing gene rank (left) and raw and covariate-adjusted (partial reg.) cell scores by linear mixed model in individuals <40 or >40 years old (right). **j**, Heatmap of enrichment of top 200 factor genes in cell subset-level DE by GSEA in cMono, ncMono and macrophages in BLO, BM, SPL, LN, LNG and JEJ. **k**, APOE–TREM2 signature cell scores from a human lung dataset (two-sided Wilcoxon rank-sum test) (left) and enrichment of the factor in macrophage DE by GSEA across age (right). For all panels, q denotes adj. P value (FDR); *adj. P value (FDR) < 0.05.

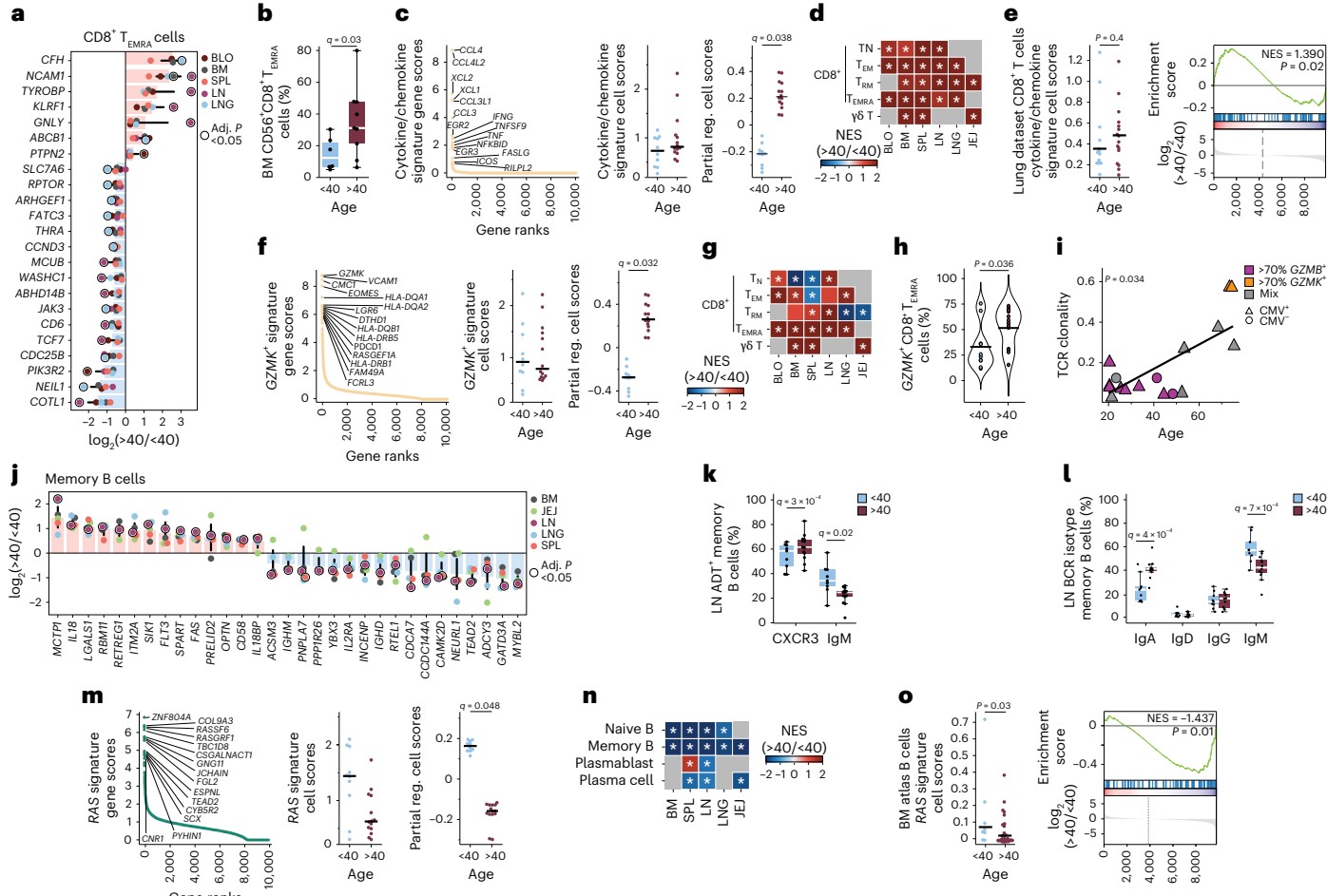

**Fig. 6 | Immune aging in adaptive lymphocytes varies across tissues. a**, Bar-and-dot plots of DEGs from individuals <40 or >40 years old in CD8[+] $T_{EMRA}$ cells in BLO, BM, SPL, LN and LNG by linear mixed model (dreamlet). Bars show median log$_2$(FC) across tissues; error bars, 95% CI; dot color indicates tissue, with dots for significant genes (adj. *P* value (FDR) < 0.05) encircled in black. **b**, Box plot (center, median; box, IQR; whiskers, 1.5× IQR) showing percentage of CD8[+] $T_{EMRA}$ cells expressing CD56 in the BM of donors <40 or >40 years old. Statistical significance determined by generalized linear model (two-sided Wald test). **c**, Consensus scHPF CD8[+] T cell cytokine/chemokine signature, with dot plots showing gene rank (left) and raw and covariate-adjusted (partial reg.) cell scores by linear mixed model in donors <40 or >40 years old (right). **d**, Heatmap of enrichment of top 200 genes in the cytokine/chemokine signature in cell subset-level DE by GSEA in CD8[+] T cell subsets from BLO, BM, SPL, LN, LNG and JEJ. **e**, CD8[+] cell cytokine/chemokine signature cell scores in CD8[+] T cells from a human lung dataset in donors <40 or >40 years old (two-sided Wilcoxon rank-sum test) (left) and enrichment of factor in CD8[+] T cells DE across age by GSEA (right). **f**, Consensus scHPF CD8[+] T cell *GZMK* signature with dot plots showing gene rank (left) and raw and covariate-adjusted (partial reg.) cell scores by linear mixed model in donors <40 or >40 years old (right). **g**, Heatmap of enrichment of top 200 genes from the *GZMK* signature in cell subset-level DE by GSEA in subsets of CD8[+] T cells in BLO, BM, SPL, LN, LNG and JEJ. **h**, Violin plot showing frequency of *GZMK*[+]CD8[+] $T_{EMRA}$

cells in BM, SPL and BLO in donors <40 or >40 years old. Statistical significance determined by two-sided Wilcoxon rank-sum test. **i**, Scatterplot of clonality score (1 − Pielou's evenness index) for CD8[+] $T_{EMRA}$ cells with >70% *GZMB*, >70% *GZMK* or a mix of *GZMB* and *GZMK* expression in BM, SPL and BLO of CMV[−] or CMV[+] donors, with statistical significance determined by linear model. **j**, Bar-and-dot plot of DEGs in donors <40 or >40 years old in memory B cells as in **a**. **k**, Box plot (center, median; box, IQR; whiskers, 1.5× IQR) showing frequency of CXCR3[+] and IgM[+] memory B cells in the LN of donors <40 or >40 years old. Statistical significance determined by generalized linear model (two-sided Wald test). **l**, Box plot of BCR showing the frequency of IgA[+], IgM[+], IgG[+] or IgD[+] memory B cells in LN of donors <40 or >40 years old with statistical significance determined by generalized linear model (two-sided Wald test). **m**, Consensus scHPF revealing RAS signature in memory B cells, with dot plots showing gene rank (left) and raw and covariate-adjusted (partial reg.) cell scores by linear mixed model in donors <40 or >40 years old (right). **n**, Heatmap of enrichment of top 200 genes in the RAS signature in cell subset-level DE by GSEA in naive B cells, memory B cells, plasmablasts and plasma cells in BM, SPL, LN, LNG and JEJ. **o**, B cell RAS signature cell score in B cells from a human BM atlas in donors <40 or >40 years old (left) and enrichment of factor in B cells DE across age by GSEA (right). For all panels, *q* denotes adj. *P* value (FDR); *adj. *P* value (FDR) < 0.05.

For B cells, there were more age-related DEGs in memory compared to naive B cells in LNs (Fig. 6j and Extended Data Fig. 5d). Memory B cells from older donors had increased expression of genes associated with inflammatory cytokines (*IL18*, *IL18BP*), cell adhesion (*CD58*, *LGALS1*) and cell death or autophagy (*FAS*, *ITM2A*), along with reduced expression of proliferation (*CDCA7*, *IL2RA*), lipid metabolism (*ACSM3*, *PNPLA7*) and differentiation markers (Fig. 6j). Select IL-18 pathway-associated genes in B cells and CD4[+] T cells in blood were validated in the Human Immune Health Atlas cohort[49] (Supplementary Fig. 9a,b). Transcript

and/or surface expression of IgM (*IGHM*) and *IGHD* were reduced in older compared to younger B cells, and the frequency of IgM[+] B cells decreased with age (Fig. 6k,l and Supplementary Table 20). We identified by scHPF that a gene signature related to RAS signaling (*RASA4B*, *RASGRF1*, *GAB2*) downstream of the BCR[53] was downregulated with age in naive and memory B cells across all sites (Fig. 6m,n). We validated this age-associated signature in BM-derived B cells from an independent dataset (*n* = 39, age 2–84 years)[54] and in PBMCs from the Human Immune Health Atlas[49] (Fig. 6o and Extended Data Fig. 5g). Pathway

analysis further revealed increased inflammation and reduced BCR signaling in LN B cells with age (Supplementary Fig. 8c). These results showed that B cells exhibited diminished signaling and functional dysregulation across tissues over age.

## Host factors and subset heterogeneity affect T cell aging

CMV infection drives immune cell alterations, including increased accumulation of $T_{EMRA}$ cells with age in blood, spleen and lungs[4,55]. We investigated the impact of CMV serostatus on cell composition and immune aging and found no significant associations with CD4$^+$ T cell, CD8$^+$ T cell or B cell frequencies (Supplementary Fig. 10a,b). Two CD8$^+$ T cell signatures were associated with CMV serostatus after regression of other covariates: the *GZMK*$^+$ signature and a *GNLY*$^+$ signature (*GNLY*, *FGFBP2* and *CX3CR1*) (Supplementary Fig. 10c–f). The *GNLY*$^+$ signature was enriched across all CD8$^+$ T cell subsets in CMV$^+$ donors, while the *GZMK*$^+$ signature was variably enriched in different sites and subsets of CMV$^+$ donors by GSEA (Supplementary Fig. 10d,f and Supplementary Tables 21 and 22). Therefore, CMV infection drives T cell gene signature changes that overlap with, but are distinct from, age-related immune alterations.

CD4$^+$ T cells are highly heterogeneous and exhibit functional and phenotypic continuums[56,57], suggesting that age effects could differentially manifest within or across subsets. We applied an annotation-independent analysis of aging in CD4$^+$ T cells in the lung, JEJ and LN, leveraging a per-cell estimation of age effects using counterfactual analysis with MrVI, which separately considers each cell and controls for covariates[10]. This analysis identified groups of cells with similar predicted age-associated changes in gene expression ('modules'), which we interrogated by DE analysis across age (Fig. 7, Extended Data Fig. 6, and Supplementary Tables 23 and 24). In the lung, a fraction of CD4$^+$ T cells (~25%, comprising $T_{EM}$ cells, $T_{RM}$ cells and $T_{EMRA}$ cells) exhibited decreased cytotoxicity (*GZMH*, *GNLY*, *GZMA*) and increased cytokine receptor (*IL18R1*, *IFNGR1*) genes with age (Fig. 7a–c and Extended Data Fig. 6a,b). Similar upregulation of cytokine responsiveness with age occurred in some CD4$^+$ T cells in blood, BM, LN and spleen, while decreased cytotoxicity was unique to the lungs (Fig. 7c,d and Extended Data Fig. 6b). CD4$^+$ T cells in the JEJ (mainly $T_{RM}$ cells) exhibited an age-related decline in $T_H$17-associated genes (*IL17A*, *IL17F*, *IL22*, *RORC*, *CCR6*)[58] and increase in pro-inflammatory cytokines (*IFNG*, *TNF*) (Fig. 7e–g and Extended Data Fig. 6c,d); age-associated downregulation of IL-17-associated genes was also observed in CD4$^+$ T cells in lung and blood (Fig. 7g–h and Extended Data Fig. 6d). CD4$^+$ T cells in the LN (also in spleen and lungs) exhibited reduced expression of genes associated with regulation (*IL10*, *TIGIT*, *CTLA4*, *CD27*) and increased expression of inflammation and activation markers (*IL18BP*, *TNFRSF4*, *TNF*) with age (Fig. 7i–l and Extended Data Fig. 6e,f). These results revealed age-associated transcriptional changes in tissue CD4$^+$ T cells associated with site-specific functions.

## Discussion

We present a comprehensive analysis of the human immune system across tissues and ages through multimodal profiling of blood and tissues from organ donors spanning six decades of adult life. We found that tissue localization was a dominant driver of the immune landscape, determining immune cell composition, cell states and functional capacity. With age, these tissue-specific properties were largely maintained, although certain subsets and sites showed altered function, migration and regulation. Our results reveal that the human immune system is highly specialized for diverse tissue environments to maintain homeostasis and mount effective immune responses.

We demonstrated that each tissue imposed site-specific immune cell compositions and adaptations that varied by lineage, and these tissue effects were conserved across donors. Although we realized and reinforced site-specific features for $T_{RM}$ cells at barrier sites and lymphoid organs[7,59], whether these adaptations applied to other immune

cells remained unknown. Here, we found that site-specific signatures for T cells in the gut (high tissue residency, low cytotoxicity), lungs (high effector function, increased regulation) and lymphoid organs (stem-like features) were not exclusive to the canonical resident populations, were shared across NK cell and ILC subsets and were absent from B cell and myeloid lineages. The enhanced expression of stem-like transcription factors TCF-1 and LEF-1 by LN memory T cells suggests that lymphoid organs may serve as reservoirs for long-lived memory cells, given these factors' requirement for memory T cell generation[59,60]. Macrophages and plasma cells also exhibited site-specific features in the gut, lungs and lymphoid organs through distinct subset-specific pathways, such as alveolar macrophages in the lungs and red pulp macrophages in the spleen. These lineage-dependent tissue adaptations probably reflect niche localization and interactions with distinct structural and immune cells within each tissue.

Age-associated gene signatures identified for macrophages, T cells and B cells were intrinsic to the subset and site. The *APOE–TREM2* gene signature, essential for crucial macrophage functions[46], was reduced with age by lung macrophages. *APOE–TREM2* expression in microglia is associated with neurodegeneration in Alzheimer's disease and in other macrophage types with cardiovascular diseases[61,62]. TREM2 can have different effects on macrophage functions; promoting anti-inflammatory 'M2-like' function in some contexts and phagocytosis and sustained inflammation in others[63,64]. The age-associated loss of TREM2 in lung macrophages could thus account for compromised immunity to respiratory pathogens and increased lung cancer susceptibility observed in the aging population. TREM agonists that enhance phagocytic function are being tested in clinical trials in Alzheimer's disease[65] and could be considered in the rejuvenation of aging macrophages in other sites.

Other age-associated features were specific to lymphocyte lineages. T cells in circulation expressed higher levels of genes associated with inflammation, cytotoxicity and NK-like markers with age, as previously reported[48,66]. Circulating $T_{EMRA}$ cells and $T_{EM}$ cells upregulated *GZMK* and other markers, similar to senescent GzmK$^+$CD8$^+$ T cells found in mice and human blood[52]. $T_{RM}$ cells in the lungs and intestines did not exhibit this age-associated gene signature, suggesting that the tissue environment may insulate them from signals that promote cellular aging or that cellular aging is tissue-specific. However, both circulating ($T_{EM}$, $T_{EMRA}$) and $T_{RM}$ cells had increased expression of genes for pro-inflammatory cytokines and chemokines with age, consistent with inflammaging implicated in cardiovascular diseases and metabolic dysregulation[67,68]. Our findings suggest that human T cells may be more prone to innate functions such as cytokine-driven activation (for example, via IL-18) with age. We also identified an age-associated increase in IL-18 expression and reduced BCR-mediated signaling within tissue B cells, which is a feature of NK-like B cell subsets identified in disease contexts[69,70]. Thus, aging may reflect a broader age-related shift to innate-like functions in both T cells and B cells.

Our findings have important implications for immune monitoring, therapeutic modulation and clinical advancement. The compartmentalization of immune subsets across tissues emphasizes the importance of site-specific immune monitoring in disease states, as exemplified in severe COVID-19, in which immune dynamics in the respiratory tract rather than blood correlated with infection outcome[71,72]. The distinctness of gut-specific subsets provides rationale for targeted intestinal interventions, as demonstrated by rotavirus vaccines[73]. The identification of stem-like profiles (marked by *TCF7* and *LEF1* expression) in LN T cells and NK cells has direct relevance to adoptive CAR-T immunotherapies, in which stemness is associated with remission[74]. LNs may thus represent an optimal source of NK and T cells for engineering adoptive cell therapies against cancer, infections and autoimmunity[75,76].

Our study has several limitations. The low frequency of certain immune subsets in tissues, including DCs, macrophages in lymphoid organs and hematopoietic progenitors, precluded aging analysis and

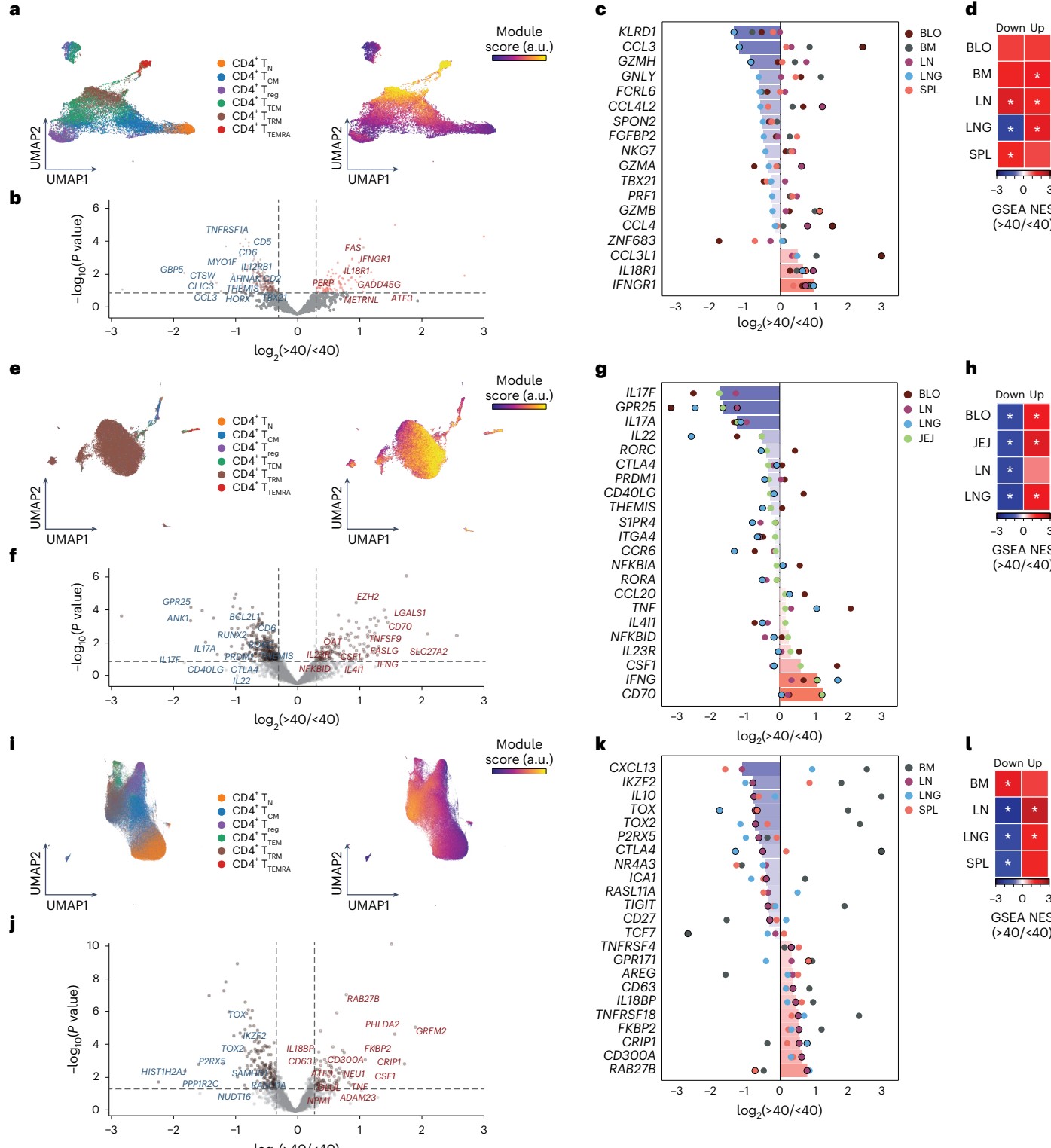

**Fig. 7 | Integrated analysis reveals tissue-dependent signatures of aging in CD4⁺ T cells. a–l,** MrVI analysis shows modules of genes changing over age for CD4⁺ T cells in the lungs (**a–d**), gut (**e–h**) and LNs (**i–l**), depicted as UMAP embeddings (**a**, **e** and **i**), colored by subset (left) and tissue module score (right). Age-associated DEGs identified by linear mixed model (dreamlet) in donors <40 or >40 years old are shown in volcano plots for module-positive cells in each tissue (**b**, **f** and **j**) and box-and-dot plots (**c**, **g** and **k**) in BLO, BM, LN, LNG and SPL (**c**), BLO, LN, LNG and JEJ (**g**), and BM, LN, LNG and SPL (**k**), and heatmaps (**d**, **h** and **l**) showing enrichment of manually selected module genes across BLO BM, LN, LNG and SPL (**d**), BLO, LN, LNG and JEJ (**h**), and BM, LN, LNG and SPL (**l**)

in tissue-specific DE of module-positive cells by age using GSEA. In **a**, **e** and **i**, module score is computed as the sum of inferred log₂(FC) for genes upregulated in donors >40 years old minus the sum of log₂(FC) for genes downregulated in individuals >40 year old. In **b**, **f** and **j**, selected genes with high effect size (by estimated log₂(FC), unadj. *P* < 0.05) are labeled. Bars show median log₂(FC) across tissues; dot color indicates tissue with trending DEGs (unadj. *P* < 0.05) outlined in black. In **c**, **g** and **k**, genes with known effector or activation functions are shown, selected from the module or trending DEG list. Tissues are included only if sufficient signature-positive cells were detected. Separate analyses (columns) were conducted for upregulated and downregulated genes. *Unadj. *P* < 0.1.

will require sorting for future studies. Similarly, an in-depth analysis of TCR and BCR across sites and age would require isolating memory T cells and B cells from each site. Finally, although we identified age-associated changes in 24 donors, additional donors would increase power and probably reveal additional aging signatures. In conclusion, this dataset, along with the models and analyses presented, can serve as a valuable and actionable resource, informing targeted immune modulation by site and age in future treatments for infectious, neoplastic and inflammatory diseases.

## Online content

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

## Methods

### Ethics statement

This study uses samples obtained from deceased organ donors and does not qualify as human subjects research in the USA, given that the donors are deceased and not living, as confirmed by the Columbia University Institutional Review Board. In the UK, samples were collected and analysed under an ethically approved research protocol (REC 15/EE/0152).

### Statistics and reproducibility

The study analyzed immune cells from multiple tissue samples obtained from 24 organ donors. No statistical method was used to predetermine sample size. No data were excluded from the analysis. Investigators were not blinded to allocation during experiments and outcome assessment, as this is a profiling study.

### Tissue acquisition from organ donors

Tissues were obtained from deceased organ donors (Supplementary Table 1) at the time of organ acquisition for clinical transplantation. In the USA, this was done through an approved protocol and material transfer agreement via LiveOnNY, the organ procurement organization for the New York metropolitan area[3,77]. In the UK, tissues were obtained through the Cambridge Biorepository for Translational Medicine (CBTM), REC 15/EE/0152, as previously described[9]. Owing to the different amounts of tissues and some distinct samples (for example, skin, liver and colon) obtained at each location, protocols for processing may differ, as described below.

### Tissue processing and CITE-seq at Columbia University

Each tissue was subjected to a tissue-specific protocol to maximize MNC recovery and viability across a diversity of sites[9]. Detailed, step-by-step protocols for immune cell isolation from blood, BM, spleen, LNs, lungs (parenchyma and airway or BAL) and JEJ (JLP and JEL) are presented elsewhere[78]. All single-cell suspensions from each site were centrifuged (400$g$, 10 min at 4 °C) and washed twice with PBS containing 5% (v/v) FBS and 2 mM EDTA. Cells were counted using the NC-2000 Cell Counter (Chemometec), and 50 million viable cells from each site were treated with TruStain FcX (BioLegend) and FcR Blocking Reagent (Miltenyi). Cells were subsequently labeled for 30 min at 4 °C with biotinylated anti-CD66B, anti-CD235ab and anti-CD326 to remove granulocytes, red blood cells and epithelial cells, respectively, by streptavidin-coated magnetic particles and negative selection (Bangs Laboratories). All single-cell suspensions were subjected to dead cell removal using a Dead Cell Removal Kit (Miltenyi).

Each single-cell suspension was hashtagged to allow pooling of samples for loading on the 10× Genomics Chromium instrument. MNCs from each site ($10^6$ per site) were transferred into 4 ml flow cytometry tubes, pelleted by centrifugation as above and resuspended in PBS containing 5% (v/v) FBS and 2 mM EDTA and then incubated with TruStain FcX (BioLegend) and FcR Blocking Reagent (Miltenyi) at 4 °C for 10 min to reduce background labeling. Each hashtag was spun at 14,000$g$ for 10 min, added to each sample (1 µl hashtag per tube), incubated at 4 °C for 30 min, pelleted and washed 3 times with PBS containing 5% (v/v) FBS and 2 mM EDTA. For CITE-seq antibody staining, 200,000 cells from each sample were resuspended in reconstituted TotalSeq-A Universal Cocktail (BioLegend) (donors 496 and 503) and TotalSeq-C Universal Cocktail (BioLegend) (remaining 10 donors) in PBS containing 5% (v/v) FBS and 2 mM EDTA, incubated at 4 °C for 30 min and washed 3 times with PBS containing 5% (v/v) FBS and 2 mM EDTA before resuspension in a final volume of 1 ml. CITE-seq antibody panels are listed in Supplementary Table 1.

### Tissue processing and CITE-seq at the University of Cambridge

Each tissue was subjected to a tissue-specific protocol to generate a single-cell suspension of immune cells that has been published in detail elsewhere[79–81]. Immune cells were isolated from blood, BM aspirates (sternum), spleen, LNs, lungs, liver, JEJ (JEL and JLP) and skin. Each single-cell suspension was hashtagged to allow pooling of samples for loading on the 10× Genomics Chromium instrument. Approximately 500,000 MNCs per tissue were transferred into 1.5 ml Lo-Bind DNA tubes. Cells were centrifuged at 400$g$ for 5 min, the supernatant removed and resuspended in 50 µl PBS containing 0.04% BSA. Cells were treated with 5 µl TruStain FcX (BioLegend) to reduce background labeling and incubated at 4 °C for 10 min, then each hashtag was added to the sample (1 µl hashtag per tube). Samples were incubated at 4 °C for 30 min, washed three times with PBS containing 0.04% BSA and equal numbers of cells from each tissue were pooled based on the number processed per donor. Cells were incubated with TotalSeq-C Human Universal Cocktail (BioLegend) (Supplementary Table 2) for 30 min at 4 °C and subsequently washed 3 times with PBS containing 0.04% BSA. Cells were resuspended in 500 µl PBS containing 0.04% BSA and passed through a 40 µm Flowmi pipette tip filter to remove any clumps of cells.

### Single-cell sequencing

For scRNA-seq experiments, single cells were loaded onto the channels of a Chromium chip (10× Genomics). cDNA synthesis, amplification and sequencing libraries were generated using either the Single Cell 5′ Reagent (v1 and v2) or 3′ Reagent (v3) Kits. TCRαβ and BCR paired VDJ libraries were prepared from samples made with the 5′ Reagent Kit. All libraries were sequenced on either an Illumina HiSeq 4000, NextSeq or NovaSeq 6000 instrument.

### Alignment and preprocessing of CITE-seq and scTCR/BCR-seq using Cell Ranger

Alignment was performed using Cell Ranger (v6.0.0) from 10× Genomics[82] with the appropriate chemistry option (*fiveprime* or *SC3Pv3*). We added the cell hashing antibody and the protein antibody fastqs to a single call of *cellranger count*. Immune receptors (TCR and BCR) were aligned using *cellranger vdj*. TCR and BCR alignment results from Cell Ranger were used for quality control and filtering of low-quality cells (individual cells with both TCR and BCR detected). In cases where a single cell had both TCR and BCR reads, the immune receptor data were discarded, and the cell was labeled as a multiplet. For all alignments, we used reference genome *refdata-gex-GRCh38-2020-A* and immune receptor reference *refdata-cellranger-vdj-GRCh38-alts-ensembl-5.0.0*.

### Quality control

Samples were demultiplexed by hashtag expression using *hashsolo* with default parameters[83]. Cells that were not uniquely assigned to an individual sample were removed from downstream analysis. Filtering was performed to remove cells with fewer than 50 unique genes detected. Mitochondrial counts were quantified, summing all genes starting with '*MT-*', and ribosomal counts were quantified using all genes starting with '*RPS*' and '*RPL*'. For erythrocyte-related counts, all genes starting with '*HB*' as well as *ALAS2* and *EPOR* (to detect erythrocyte precursors) were quantified. Cells with more than 20% mitochondrial counts were flagged as potentially low-quality for later filtering. Counts for mitochondrial genes and *MALAT1* were subsequently removed from the gene expression object and downstream analysis. To exclude contamination from ambient RNA, we processed the data using DecontX[84]. Two samples (one from liver and one from skin) with abnormally high ambient counts were removed, as DecontX could not correct the ambient counts (for example, plasma cell genes like *ALB* in all immune cells). All downstream analysis was performed on uncorrected counts, as we found few ambient counts in other samples. We used a CellTypist model[9] (at https://cog.sanger.ac.uk/celltypist/models/Red_Blood_CZI/v1/Red_Blood_CZI.pkl) to detect erythrocytes. Doublets were additionally detected using Scrublet[85] with a *sim_doublet_ratio* of ten. For each unique tissue site, we performed an initial integration across all samples by training a single-cell variational inference (scVI) model[86] on the gene

expression with following parameters: 10,000 highly variable genes using the *seurat_v3* option[87] in Scanpy, early stopping enabled and 50 epochs, 10 epochs for *n_epochs_kl_warmup*, two layers in encoder and decoder, *nb* gene likelihood and a mini-batch size of 256.

To perform filtering of low-quality events, we used the following quality metrics: the probability of a doublet predicted by Scrublet, the probability of a doublet from HashSolo, the percentage of erythrocyte genes as described above, whether a cell contained both TCR and BCR, whether the CellTypist erythrocyte model predicted a cell to be an erythrocyte, as well as cells with a total count below 2,000 unique molecular identifiers, 1,200 unique genes or 200 protein counts. All scores were added to generate a per-cell quality metric. To perform filtering, we argued that cells that group together and have evidence of low quality should be removed from downstream analysis. We first used Louvain clustering[88] on the coordinates from scVI latent space using 15 nearest neighbors to cluster the per-tissue integrated data with a resolution of 5.0. Every cluster with a median low-quality score (described above) of at least one was removed from downstream analysis. Although some low-quality events were retained with this filtering, their frequency was drastically reduced. We additionally established tissue-specific cut-offs to remove additional events and removed clusters with a mean low-quality score of 0.3 from all tissues, except for the lung LN and JEJ, for which the threshold was manually increased to recover higher-quality cells. Using a course cell-type annotation based on manual annotation of clusters, we identified cell types that were consistently filtered out, even though their quality did not appear to be spuriously low by manual inspection. We retained mast cells and hematopoietic stem cells from all tissues, all macrophages from LNs and spleen, all erythrocytes and platelets from BM and all monocytes from liver.

We concatenated cells from all tissues and computed the 10,000 top highly variable genes using the *seurat_v3* option in Scanpy and used the same parameters as described above, but with a mini-batch size of 1,024 to accelerate the training process. We used this integrated latent space to assign initial cell types and removed all cell types that were not labeled as immune cells. Additionally, we removed all cells for which manual labeling and automatic labeling using MMoCHi (see below) were inconclusive about coarse cell-type identity (for example, B cell, myeloid, T cell). These events were of low quality by manual inspection. We performed post hoc manual removal of these and other clusters of low-quality cells after integrating all cells.

### Immune cell subset classification using MMoCHi

To identify canonical immune cell subsets, we used a recently reported, supervised machine learning algorithm, MMoCHi) (v0.2.1)[12]. We first normalized the gene expression (GEX) count matrix using $\log(10,000C_{g,i} / T_{G,i} + 1)$, where $C_{g,i}$ represents the counts for GEX feature $g$ in cell $i$, and $T_{G,i}$ is the total counts for all GEX features in cell $i$. Similarly, we normalized the antibody-derived tag (ADT) count matrix using $\log(1,000C_{a,i} / T_{A,i} + 1)$, where $C_{a,i}$ represents the counts for ADT feature $a$ in cell $i$ and $T_{A,i}$ is the total counts for all ADT features in cell $i$. We applied landmark registration (MMoCHi) to batch-correct the ADT expression across experimental batches. In brief, we applied automatic detection of landmarks (peaks) in the expression distributions for each ADT feature in a given sample, applying manual adjustments as needed using the graphical user interface, then performed curve registration and warping to align the positive and negative peaks for each ADT feature across batches.

We provided MMoCHi with a hierarchy of immune cell subsets and their canonical surface protein-level and RNA-level markers (Supplementary Fig. 2) and used the markers to identify high-confidence members (cells) of each subset for training. For each classification level, automatic thresholds for high-confidence ADT or GEX marker-positive and marker-negative cells were manually adjusted as needed using the supplied GUI (Supplementary Table 3). Following MMoCHi's internal training data refinement, we applied an 80–20 train–test split and

trained a random forest classifier, *sklearn.ensemble.RandomForestClassifier*, on both gene and protein expression[12]. For 2 of the 24 organ donors and a subset of samples from a third donor, we did not perform CITE-seq and only had scRNA-seq profiles. Thus, these samples were excluded from the MMoCHi classification described here. However, we used a *k*-nearest neighbors approach to transfer the classifier labels to individual cells profiled from these two organ donors. Specifically, we used the *sklearn.neighbors.KNeighborsClassifier* with *n_neighbors* = 10 to construct a *k*-nearest neighbors graph in the mrVI embedding of the dataset (see below) and classify the remaining cells. Of the subsets, pDCs were identified using two separate nodes on the hierarchy (Supplementary Fig. 1), as pDCs shared expression with both B cells and myeloid cells. Once classified, the two subsets were merged into a single population of pDCs. The MMoCHi annotation was used at two separate levels throughout the paper, defined as either one of the 34 fine-grained subsets or grouped into CD4+ T cells, CD8+ and unconventional T cells (including γδ T cells and CD8+ MAIT cells), B cells, NK cells and ILCs or myeloid cells (including monocytes, macrophages, cDCs, migratory DCs and pDCs).

### Training and application of label transfer models

Owing to the breadth of tissues and human subjects sampled in our dataset and high-resolution annotation of immune subsets, we anticipated that our immune atlas would be useful to the research community as a reference for performing cell-type label transfer. To facilitate this application, we trained a model using popV[22], a tool developed for cell-type label transfer that uses several annotation algorithms and consensus voting to determine annotations and evaluate their confidence. popV also calculates joint embeddings of the query and reference datasets, which can be used for visualization of the query data and other analysis tasks. A popV model was trained using the tissues and MMoCHi annotations (Fig. 2) as the reference dataset. Label transfer performance was evaluated using the Human Lung Cell Atlas as a query dataset[89]. To visualize the data, we computed UMAP embeddings as described above on joint scVI embeddings, which were calculated as part of the popV pipeline.

To evaluate the importance of ADT information in the MMoCHi classification performance, we additionally applied a pre-trained CellTypist[9] model (Immune_All_Low; https://celltypist.cog.sanger.ac.uk/models/Pan_Immune_CellTypist/v2/Immune_All_Low.pkl) using default settings to the tissue immune cells (Fig. 2).

### Integration and cell state embedding using MrVI

To integrate scRNA-seq profiles of immune cells in our study, we first used scVI, which did not yield a fully integrated latent space and clustered by site of collection (for example, US or UK). We next leveraged MrVI, which uses a mixture-of-Gaussian as a prior and enforces stronger separation of true cell state and effect of donors on gene expression, as has been recently demonstrated[10]. MrVI takes advantage of a prior based on a multimodal variational mixture of posteriors (similar to a VampPrior[90]), which have been shown to outperform Gaussian priors for scRNA-seq integration in benchmarking studies[90]. In brief, MrVI finds a sample-agnostic latent space, $U$, and computes a sample-specific embedding. A second latent space, $Z$, is defined by adding an attention-based concatenation between $U$ and the sample embedding space to the original $U$-space. Another layer of attention is used to incorporate an embedding of 10× Genomics chemistry and experimental site (Cambridge, UK versus Columbia, NY), and this third latent space is decoded using a linear decoder to yield the rate of a negative binomial distribution. We use a cell-type-aware Gaussian mixture prior in $U$-space. To introduce cell type awareness, we use a bias to the mixture proportions that makes it likely for cells of the same type to be sampled from the same Gaussian.

For the latent embedding highlighted throughout the article and used for manual cell-type curation, we used the donor identities

as the sample keys and used the output of MMoCHi classification (see above) as the cell-type prior in MrVI. We used default parameters except *n_epochs_kl_warmup* of 25, *n_latent_u* of 20, *n_latent* in *Z*-space of 200, dropout in *qz* as well as *pz* of 0.03 (adopted from a previous publication[10]). To visualize cells (either the total immune component or individual major lineages), we computed nearest neighbors (*scanpy.pp.neighbors*) on the MrVI U latent space and calculated UMAP embeddings (*scanpy.tl.umap*) using the 15 nearest neighbors, a minimum distance of 0.4, a spread of 1.0 and initialization with PAGA after running *scanpy.tl.paga*. To identify additional heterogeneity in cell states within samples in addition to the cell-type annotation provided by MMoCHi, we performed manual annotation. For each MMoCHi annotated population, a new scVI model was trained with donor as the batch key, then Leiden clustering (*scanpy.tl.leiden*) was performed on the lineage-specific neighbors graph at an appropriate resolution, selected to minimize over-clustering (ranging from 1 to 15). Markers for each cluster were computed by *scanpy.tl.rank_genes_groups*, and clusters with similar marker expression were merged. To annotate proliferating cells, *scanpy.tl.score_genes_cell_cycle* was run, and the output was used in combination with the gene expression of *MKI67* and *TOP2A*.

## DE and variance decomposition using dreamlet

We focused our DE analysis on immune lineages and cell types with sufficient representation across experimental sites, tissues and donor ages. This included six tissue groups: blood, BM, spleen, gut (JLP and JEL), LNs (ILN, LLN and MLN) and lungs (consisting of BAL and parenchyma); six immune lineages: myeloid, CD4+ T cells, CD8+ T cells, invariant T cells (that is, γδ T cells and MAIT cells), B cells and ILC/NK cells; and 26 individual cell types within all lineages. Covariates included 10× Genomics chemistry (3′ versus 5′), sex (male versus female), laboratory (Cambridge, UK versus Columbia, NY) and CMV status (positive versus negative). For aging analyses, donors were categorized as being <40 or >40 years of age.

Variance decomposition and pseudobulk DE analysis were performed using LMM through the dreamlet R package (v1.4.1)[23]. Depending on the resolution of the analysis, DE was performed separately either for each immune lineage (for example, myeloid cells, B cells, and so on) or for each immune subset (for example, macrophages, naive B cells and so on) using the *cluster_id* parameter in *dreamlet*. The raw GEX count matrix was pseudobulked across samples, and each tissue in each donor was treated as a separate sample. Before performing DE, samples and genes with poor representation were filtered using *dreamlet::processAssays*. Samples with fewer than 50 cells and genes not represented in at least 40% of the samples with at least five counts were excluded. To confirm findings by MrVI counterfactual analysis (see below), these thresholds were reduced to a minimum of ten cells for a sample to be included, and at least 10% of samples with at least five counts. DE for a subset was not performed when fewer than three or four samples (for tissue and age analysis, respectively) met the minimum cell thresholds. Variance decomposition was performed for age analysis for each lineage using *dreamlet::fitVarPart* with sex, sequencing chemistry, CMV serostatus, age group, processing site and tissue as covariates (Supplementary Table 13). LMM was performed using *dreamet::dreamlet* with eBayes estimation enabled. Tissue effects (Figs. 3 and 4, Extended Data Figs. 3 and 4, and Supplementary Tables 4, 7 and 10) were modeled by comparing each lineage/subset in one tissue group to the same lineage/subset in the remaining tissue groups, with donor identity encoded as a random effect. Age effects (Figs. 5–7, Extended Data Fig. 5, and Supplementary Tables 15 and 24) were modeled across each tissue-group–age-group combination while controlling for CMV serostatus and sex as fixed effects and with sequencing chemistry and processing site as random effects. Age effects within each tissue group were then measured using the *contrasts* parameter in *dreamlet::dreamlet* between old and young for each tissue group (for example, the effect of age in the gut was computed as 'old-gut – young-gut'). CMV effects (Supplementary Fig. 10 and Supplementary Table 21) were modeled across each tissue-group–CMV serostatus combination while controlling for age and sex as fixed effects and with sequencing chemistry and processing site as random effects. CMV effects within each tissue group were then measured using the *contrasts* parameter in *dreamlet::dreamlet* between CMV+ and CMV− for each tissue group.

## Identification of gene co-expression patterns using consensus scHPF

For identifying cross-tissue and cross-donor gene signatures for each major immune lineage, we constructed probabilistic factor models directly from scRNA-seq count matrices using scHPF. The output of scHPF includes two matrices: an *M* × *K* gene score matrix containing weights for each of *M* genes in each of *K* factors and a *K* × *N* cell score matrix containing weights for each of *N* cells in each of *K* factors. In the original report of scHPF, the algorithm required a user-supplied value of *K*, the number of factors in the model[91]. Here, we use a new consensus factorization implementation of scHPF, in which the user specifies a broad range of *K* values from which many scHPF models are generated[27]. The gene score matrices for these models are then clustered to identify *K* recurrent factors, which are combined to seed a final round of training to construct a final consensus model with *K* factors.

We constructed two types of scHPF models: a tissue-level model (Extended Data Fig. 4), in which the number of cells from each of three tissue groups was balanced by random sub-sampling (gut: JEL and JLP; lung parenchyma; and LNs: MLN and LLN), and a donor-level model (Figs. 5 and 6), in which the number of cells from each organ donor was balanced. We constructed both types of models for the major immune lineages: CD4+ T cells, CD8+ T cells (including all invariant T cells), NK cells, ILCs, B cells and macrophages. For donor models, donors with fewer than 300 cells for a given lineage were removed. In both models, the count matrices were randomly downsampled such that the average number of transcripts per cell was the same for each donor to avoid coverage bias. scHPF models considered only protein-coding genes (excluding TCR and immunoglobulin cassettes) detected in at least 1% of cells across the final subsampled and downsampled training matrix.

For all consensus scHPF models, we ran scHPF five times for each of 16 values of *K* (15–30), from which we selected the top three models for each value of *K* based on convergence criteria for clustering. We applied walktrap clustering to identify recurrent clusters, which we required to form clusters with factors from at least two different models from which we trained the final consensus model[27].

## Detection of tissue-specific effects

Immune subset composition within each lineage across tissues was visualized by violin plots or box plots (using *seaborn*). Tissue-specific enrichment of immune subset frequencies in specific tissues was also assayed within each major lineage using scCODA[92] for Bayesian inference. Significant enrichment of an immune subset in one tissue over the rest was determined using *sccoda.util.comp_ana.CompositionalAnalysis* to detect credible effects, and was run sequentially, selecting each cell type as the reference. Majority voting was then used to identify cell types that are credibly changing more than half the time with automatic reference-subset selection and the default false discovery rate of 0.05.

To determine tissue-specific gene expression signatures across immune lineages (Fig. 3), significant DEGs were defined as adjusted *P* < 0.05 and log2(FC) > 1 by pseudobulk DE across tissues at the lineage level (see above). Mean *z*-score gene expression was calculated for each pairing of tissue group and lineage. Genes and samples were both hierarchically clustered using *scipy.cluster.hierarchy.linkage* with Ward's method and Euclidean distance. Discrete clusters of genes with similar expression patterns were calculated using *scipy.cluster.hierarchy.fcluster* with the '*maxclust*' method (Supplementary Table 6).

For each gene cluster, association with specific tissue groups or lineages could arise from DE within one or more specific subsets of that lineage or from compositional shifts in subsets across tissues. To disentangle these possibilities, we first used pre-ranked GSEA to compare the gene clusters identified via lineage-level DE to the effect size (that is, log(FC)) of DE across tissues in the subset-level DE. To visualize potential effects caused by compositional shifts across tissues, we computed the average frequency of the subset (as a proportion of the total cells within that lineage group) within a tissue, the FC of that frequency over the frequency in the remaining tissue groups and the average expression of the gene cluster.

To assess whether differential transcript expression was reflected in the surface protein profiling (Supplementary Tables 11 and 12), we selected ADTs corresponding to DEGs in at least one tissue. To identify enrichment in one tissue group over the other tissue groups, we used *scanpy.tl.rank_genes_groups* on the normalized expression with Wilcoxon and tie-correction enabled. To minimize the influence of technical staining artifacts or donor covariates, analysis was conducted separately within each donor. Donors with fewer than 50 cells for a particular tissue-group–lineage combination or tissue-group–lineage combinations with fewer than four suitable donors were excluded from analysis. Before DE analysis, the ADT count matrix was subsampled to equalize cell numbers and randomly downsampled such that the average number of transcripts per cell was the same for each group to avoid coverage bias.

We next sought to identify factors from the tissue-level scHPF models of each major immune lineage that were shared across cell types. As described above, we first constructed consensus scHPF models for CD4[+] T cells, CD8[+] T cells, macrophages, NK cells, ILCs and B cells with equal representation of cells from each of three major tissue groups (gut, lung and LNs). From each model, we removed probable nuisance factors containing heat shock protein-encoding genes (common dissociation artifact, >1 gene), ribosomal protein-encoding genes (common coverage artifact, >10 genes), genes from the highly inducible metallothionein cluster (>1 gene), hemoglobin transcripts (red blood cell contamination, >0 genes) and genes in a previously published signature of dissociation-induced cell stress in scRNA-seq (>7 genes)[93] among the 30 top-weighted genes. Next, we computed the average cell score for each factor in each of the three major tissue groups and identified all factors with an average tissue-group score that was at least 80% higher in one tissue group than the average of the remaining two. Thus, the resulting set of 53 scHPF factors from across all 6 lineage-specific models exhibits some degree of tissue specificity. To compare these factors to each other, we computed the Pearson correlation between the gene score vectors for each pair of factors. We then identified factors with a pairwise correlation that was greater than the 95% confidence threshold with at least two other factors, which yielded 31 scHPF factors from across the six major immune lineages. Finally, we performed hierarchical clustering of the Pearson correlation matrix for these 31 factors (*seaborn. clustermap* using Euclidean distances) to identify modules containing factors with similar gene signatures that originated from different, lineage-specific scHPF models (Extended Data Fig. 4). Modules of genes were further interrogated by average gene expression and validated in specific immune subsets using pseudobulk GEX DE and ADT DE as described above.

## Detection of age-specific and CMV effects on composition and gene expression

To detect shifts in the subset composition of specific lineages across the age groups and CMV serostatus (Supplementary Table 14), we performed generalized linear modeling by fitting a *statsmodels.GLM* model for each tissue subset, considering sex, sequencing chemistry, CMV serostatus and processing site as additional covariates. Donors with fewer than 50 cells for a particular tissue-group–lineage combination

or tissue-group–lineage combinations with fewer than four suitable donors were excluded from analysis. The estimated coefficients were used to calculate a covariate-aware $\log_2(FC)$ for visualization. We used the *statsmodels.multipletests* function to adjust $P$ values for multiple comparisons (Benjamini–Hochberg method), and subset–tissue combinations with adjusted $P < 0.05$ were considered significantly changing across age.

To depict age-associated effects on the immune system, we visualized the similarity of trending DEGs by age on immune subsets across tissues using *t*-distributed stochastic neighbor embedding (*t*-SNE) (Fig. 5d). We first calculated trending DEG (unadjusted $P < 0.05$; <40 or >40 years old) pairwise similarities by summing the intersection of positively regulated genes ($\log_2(FC) > 0.1$) and negatively regulated genes ($\log_2(FC) < -0.1$), divided by the overall union of both. The similarity or distance (1 − similarity) was applied to cell types containing more than 70 DEGs (unadjusted $P < 0.05$; mean log-normalized expression, >0.05) and present in at least 3 donors per tissue and age group. The similarity levels of cell types and tissues with more than 200 DEGs were further clustered using the Ward.D2 method and projected into a distance-based *t*-SNE illustration.

To investigate the effect of age on specific genes within each immune subset, we plotted genes that were significant in at least one tissue (adjusted $P < 0.05$) and within the top 50 significant genes. Although our power to detect age effects by DE was limited, genes that were significantly DE in one subset were often trending in the same direction across multiple tissue groups. To assess the effect of age on surface protein expression (Supplementary Table 17), we used landmark-registered protein expression data (by MMoCHi, see above) to account for donor-to-donor batch effects in ADT staining quality. Although landmark registration preserves the separation between positive-expressing and negative-expressing cells for thresholding, this non-parametric normalization can obscure changes in overall expression intensity between samples. Therefore, we focused on shifts in percent positivity for a marker in each tissue subset. We performed automatic thresholding by MMoCHi, followed by manual adjustment (as described above), on the landmark-registered expression of all ADTs corresponding to a DEG by age. The percentage of cells with expression of a given ADT above the positive threshold was calculated for each donor–tissue-group–subset combination. Donors with fewer than 50 cells for a particular tissue-group–lineage combination or tissue-group–lineage combinations with fewer than four suitable donors were excluded from analysis. The percent positivity was used as the response variable in the same linear regression model used to detect shifts in composition across age groups. We adjusted $P$ values for multiple comparisons as above, and ADTs with adjusted $P < 0.05$ were considered significant.

We constructed donor-level scHPF models for each major immune lineage with uniform representation of cells from each donor to identify age-associated gene signatures, as described above. For each scHPF model, we performed LMM to account for covariates and identify age associations. Each LMM contained six categorical covariates as fixed effects: age group, sequencing chemistry, sex, processing site and CMV serostatus. We also considered three tissue types: mucosal (BAL, lung parenchyma, JLP, JEL), LNs (ILN, MLN and LLN) and blood-rich, including blood, BM and spleen), which required us to select one category (blood-rich) as a held-out variable. Thus, we have two categorical variables for tissue, which effectively represent mucosal versus blood-rich and LN versus blood-rich. We encoded donor identity as a random effect. LMM coefficients and $P$ values were computed for each factor in a given scHPF model using the cell scores as response variables by fitting a *statsmodels.MixedLM* model and using the *statsmodels. multipletests* function to adjust $P$ values for multiple comparisons (Benjamini–Hochberg method).

To cross-validate age-associated scHPF factors in other datasets, we further analyzed a bone marrow atlas[54] containing 36 age-annotated

donors with good B cell representation for a B cell aging factor, a lung atlas[47] containing 29 age-annotated donors with good macrophage and CD8[+] T cell coverage and PBMC data from the Sound Life cohort (age 25–65 years, n = 96) from the Human Immune Health Atlas[49] (Figs. 5 and 6 and Extended Data Fig. 5). Using the published cell-type annotations from each atlas, we extracted the appropriate scRNA-seq profiles and projected them into the corresponding donor-level scHPF models generated from the data reported here using the scHPF *project* function. This resulted in cell scores for cells from the external data sets for the same factors that were generated from this data set, allowing us to compare the average cell scores for young versus older donors from the external data. As an orthogonal approach, we also performed pseudobulk DE analysis between older and younger donors (using an age cutoff of 40 years) from the external data sets, ranked the genes by FC and used GSEA to analyze the statistical enrichment of age-associated factors among young versus old donors (Supplementary Table 19). We used the top 200 genes (ranked by scHPF gene score) in each age-associated factor as gene sets for GSEA.

## Counterfactual analysis to detect age associations with MrVI

For the DE analysis described in Fig. 7, we subsetted the MrVI sample embeddings to each tissue group and modeled the predicted ε in MrVI by a linear model adjusting for covariates in sex, CMV serostatus and age group as fixed effects, considering sequencing chemistry and processing site as site covariates in MrVI. A ridge regression parameter of 0.1, owing to collinearity of cofactors, was added. This decomposition of ε was performed for every single cell. This yields an estimated effect in Z-space for each covariate. The effect vector was added to the mean cell embedding in Z-space, and DEGs were computed based on the modified and mean embedding for each cell. For downstream analysis, this matrix of estimated log(FC) for each cell and gene was further processed for each immune subset. First, all cells that were represented only in fewer than three samples were filtered out[10]. Second, for each cell type, we excluded genes with less than 0.01 raw average expression or an estimated log(FC) across age groups with a 95th percentile below 0.1, retaining only genes that might be affected by age in a group of cells. To dissect predicted gene effects into modules, neighborhood smoothing was performed using 15 nearest neighbors in U-space and multiplying two times the normalized affinity matrix by the predicted gene effects. Spectral co-clustering was performed with four gene clusters and four cell clusters, with mini-batch enabled using *sklearn.cluster.SpectralCoclustering*. Marker genes for each module were identified by averaging the predicted log(FC) across all cells from the corresponding cell module, and the top 50 genes for each module were identified. We used decoupleR-py to compute a module score of log(FC) scores using weighted means of the signs of those marker genes[94] (Supplementary Table 23).

For the lung, we isolated a gene module in CD4[+] T cells that contained $T_{RM}$ cells. To detect similar cells in other tissues, we computed the best cutoff for the module score to identify cells in a specific cell module based on Youden's J statistic, computed the module score for all cells from other tissues as described above and applied the same cutoff to all other tissues as the tissue of interest. Given that the gut contained $T_{RM}$ cells with a $T_H17$ phenotype and all other tissues had no module-positive cells, we selected all cells with a MrVI predicted negative log₂(FC) of *IL17A* below −0.05. To confirm our findings on a per-gene level, we selected module-positive cells and used pseudobulk estimates of DE using *dreamlet* (Supplementary Table 24). Samples with fewer than 5 module-positive cells or 1,000 total counts and genes with fewer than 3 total counts were removed. Aging DE was performed using the contrasts method, as described above. Genes within a shared functional group were manually selected from the MrVI signature for visualization. Pseudobulk DE analysis was performed on the classifier-predicted cells in other tissues using the same settings as above in this cell subset. Enrichment of module or selected marker genes in the pseudobulk DE

analysis was performed using GSEA implemented in *decoupler.run_gsea* (Supplementary Table 23).

## TCR and BCR repertoire analysis using Dandelion

Cell Ranger-mapped TCR and BCR contigs contained in 'all_contigs. fasta' and 'all_contig_annotations.csv' output files were re-annotated using the Dandelion preprocessing pipeline[95]. This pipeline includes the following steps: (1) sample suffix or prefix assignment to each sample barcode; (2) re-annotation of contigs with IgBLAST (v1.19.0)[96] against IMGT (international ImMunoGeneTics) reference sequences (last downloaded on 24/04/2023); (3) re-annotation of D and J genes separately using blastn to enable the annotation of contigs without the V gene present; and (4) identification and recovery of nonoverlapping individual J gene segments. For BCRs, three additional steps were also performed: (1) additional re-annotation of heavy-chain constant (C) region calls using blastn (v2.13.0) against curated sequences from CH1 regions of respective isotype class; (2) heavy-chain V gene allele correction using TIgGER (v1.0.1)[97]; and (3) BCR mutation calling. Cell-level quality control was performed using Dandelion's 'filter_contigs' function, which only considers productive VDJ contigs, asserts that a single cell should only have one VDJ and one VJ pair or only an orphan VDJ chain and explicitly removes contigs that fail these checks (except for IgM/IgD and TRB/TRD extra pairs). Contigs that did not match any cell barcodes in the gene expression data were also removed at this step. TCRs and BCRs were then grouped into clones or clonotypes. The following default sequential criteria, which apply to both chain contigs, were applied: (1) identical V and J genes usage; (2) identical junctional CDR3 amino acid length; and (3) CDR3 sequence similarity: 100% nucleotide sequence identity at the CDR3 junction for TCRs and 85% amino acid sequence similarity (based on Hamming distance) for BCRs.

TCR or BCR data were then transferred into the corresponding AnnData object. Cells without receptor data or that presented more than one receptor were discarded from further immunoreceptor analysis. For T cell analysis, cells annotated as MAIT cells or γδ T cells were also discarded. Clonality of the different populations was calculated as 1 − Pielou's evenness index, varying from zero (more diverse) to one (less diverse), with the Pielou's evenness corresponding $H_s / H_{max}$, where $H_s$ is the Shannon entropy of sample s and $H_{max} = \log_2 C$, where C is the number of unique clonotypes in s. All clonality scores were calculated on a subsample of 100 cells for each donor, cell type, tissue or cell type and tissue. To detect shifts in the BCR isotype composition of specific B cell lineages across the age groups (Supplementary Table 20), we performed generalized linear modeling by fitting a *statsmodels.GLM* model for each tissue subset, considering sex, sequencing chemistry, CMV serostatus and processing site as additional covariates. Donors with fewer than 50 cells for a particular tissue-group–lineage combination or tissue-group–lineage combinations with fewer than for suitable donors were excluded from analysis. The estimated coefficients were used to calculate a covariate-aware log₂(FC) for visualization. We used the *statsmodels.multipletests* function to adjust P values for multiple comparisons (Benjamini–Hochberg method), and subset–tissue combinations with adjusted P < 0.05 were considered significantly changing across age.

## Reporting summary

Further information on research design is available in the Nature Portfolio Reporting Summary linked to this article.

## Data availability

The global, T cell, NK/ILC, B cell and myeloid cell h5ad datasets are available on Lattice: Human Cell Atlas at https://cellxgene.cziscience.com/collections/cc431242-35ea-41e1-a100-41e0dec2665b. Raw sequencing (Fastq) files are available from the Sequence Read Archive (SRA) under accession no. SRP559768 and BioProject accession no. PRJNA1215450. Data are also available on Gene Expression Omnibus under accession number GSE299043. Source data are provided with this paper.

## Code availability

The code for data processing and downstream analysis is available on the GitHub repository at https://github.com/YosefLab/CZI-Immuneaging.

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

## Acknowledgements

We thank the deceased organ donors and their families, the extended Cambridge Biorepository for Translational Medicine (CBTM) team, the transplant coordinators at LiveOnNY for providing access to tissue samples, and J. Zamanian of the Lattice: Human Cell Atlas team for data collation and availability on CELL×GENE. This work was supported by a Chan–Zuckerberg Initiative Seed Networks for the Human Cell Atlas grant (CZF2019-002452) to N.Y., D.L.F., P.A. Sims, J.L.J. and S.A.T., along with NIH grants AI106697 and AI128949 awarded to D.L.F. and P.A. Sims and NIHR Cambridge Biomedical Research Centre funding to J.L.J. (BRC-1215-20014). D.P.C. was supported by the Columbia University Graduate Training Program in Microbiology and Immunology (T32AI106711). Flow cytometry analysis was performed in the CCTI Flow Cytometry Core, supported by NIH S10RR027050 and S10OD020056 and the Cambridge NIHR BRC Cell Phenotyping Hub (Department of Medicine, University of Cambridge). This research was funded in part by the Wellcome Trust (203151/Z/16/Z, 203151/A/16/Z) and the UKRI Medical Research Council (MC_PC_17230). For the purpose of open access, the author has applied a CC BY public copyright license to any Author Accepted Manuscript version arising from this submission. The views expressed here are those of the author(s) and not necessarily those of the NIHR, Department of Health and Social Care or National Institutes of Health (NIH).

## Author contributions

N.Y., D.L.F., P.A. Sims, J.L.J. and S.A.T. designed the study. M.K., R.M., K.M. and K.S.-P. obtained tissues from deceased organ donors. S.B.W., D.B.R., P.A. Szabo, D.P.C., S.K.H., L.B.J., K.L.E. and D.C. performed tissue dissociation and prepared single-cell sequencing libraries. M.M., C.E., S.B.W., D.B.R., P.A. Sims, P.A. Szabo, D.P.C., A.R.M., E.R. and V.V.P.A. analyzed and interpreted the data. D.L.F., N.Y., P.A. Sims and J.L.J. supervised experiments. N.Y. and P.A. Sims coordinated data analysis. D.L.F., N.Y., P.A. Sims, J.L.J., S.A.T., S.B.W., D.B.R., M.M., P.A. Szabo, C.E. and A.R.M. wrote and edited the paper. N.Y., D.L.F., P.A. Sims, J.L.J. and S.A.T. provided funding. All authors read, provided input on and approved the paper.

## Competing interests

In the past 3 years, J.L.J. has consulted for or been a member of a scientific advisory board for Sanofi, Roche and Enhanc3DGenomics, and S.A.T. S.A.T. is a scientific advisory board member of ForeSite Labs, OMass Therapeutics, Qiagen, Xaira Therapeutics, a co-founder and equity holder of TransitionBio and Ensocell Therapeutics, a non-executive director of 10x Genomics, and a part-time employee of GlaxoSmithKline. The other authors declare no competing interests.

## Additional information

**Extended data** is available for this paper at https://doi.org/10.1038/s41590-025-02241-4.

**Correspondence and requests for materials** should be addressed to Sarah A. Teichmann, Donna L. Farber, Peter A. Sims, Joanne L. Jones or Nir Yosef.

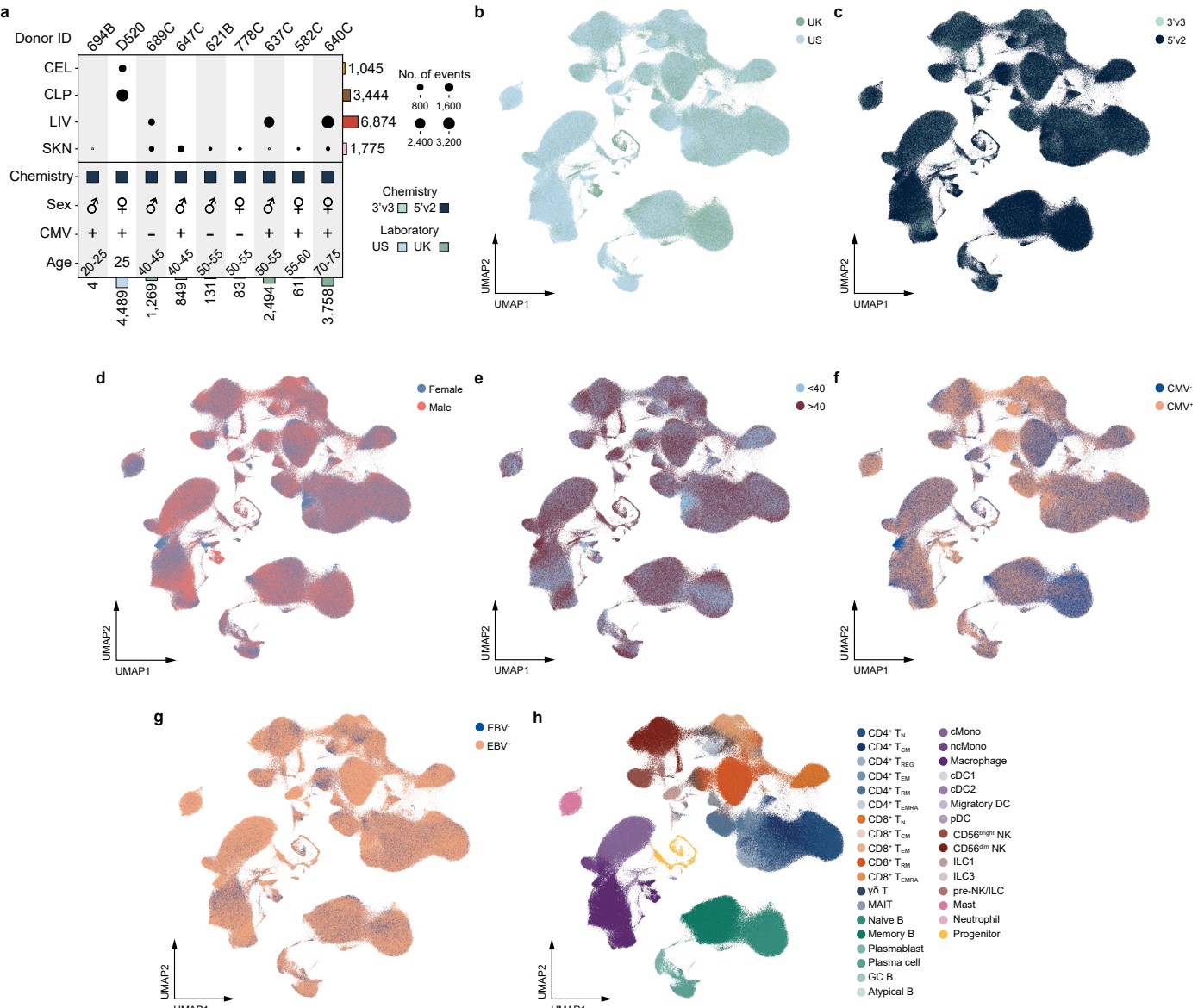

**Extended Data Fig. 1 | Additional tissues obtained and annotated covariates in the dataset. a)** Plot of cell numbers and donor metadata from tissue sites collected but excluded from analysis. **b-h)** UMAP embeddings of the dataset colored by (**b**) site of collection, (**c**) 10x Genomics sequencing chemistry, (**d**) sex, (**e**) age, (**f**) CMV serostatus, (**g**) EBV serostatus, (**h**) MMoCHi immune subset classification.

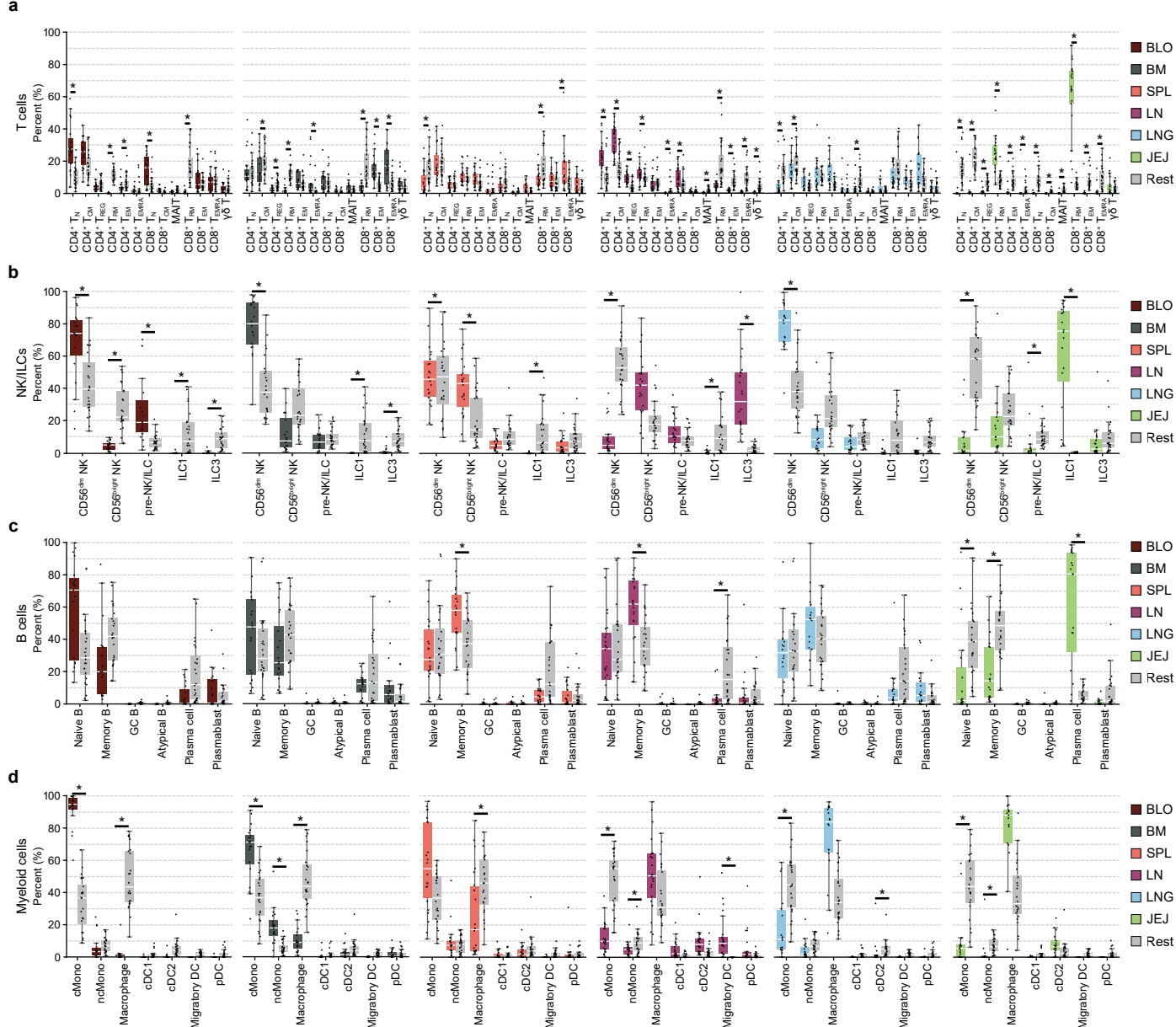

**Extended Data Fig. 2 | Immune cell subset composition varies by tissue site.**
**a-d)** Box plots showing the percentage of each immune cell subset within a tissue (colored box) compared to the average proportion across all other tissues (gray box, "Rest") for (**a**) T cells, (**b**) NK/ILCs, (**c**) B cells, and (**d**) myeloid cells. Box plots show the median (center), interquartile range (IQR; box), and whiskers extending to 1.5×IQR. Dots represent individual donors. Statistical significance determined by Bayesian modeling using scCODA. * denotes adj. p-val (FDR) < 0.05. Abbreviations: BLO (blood), BM (bone marrow), SPL (spleen), LN (lymph node), LNG (lung), JEJ (jejunum).

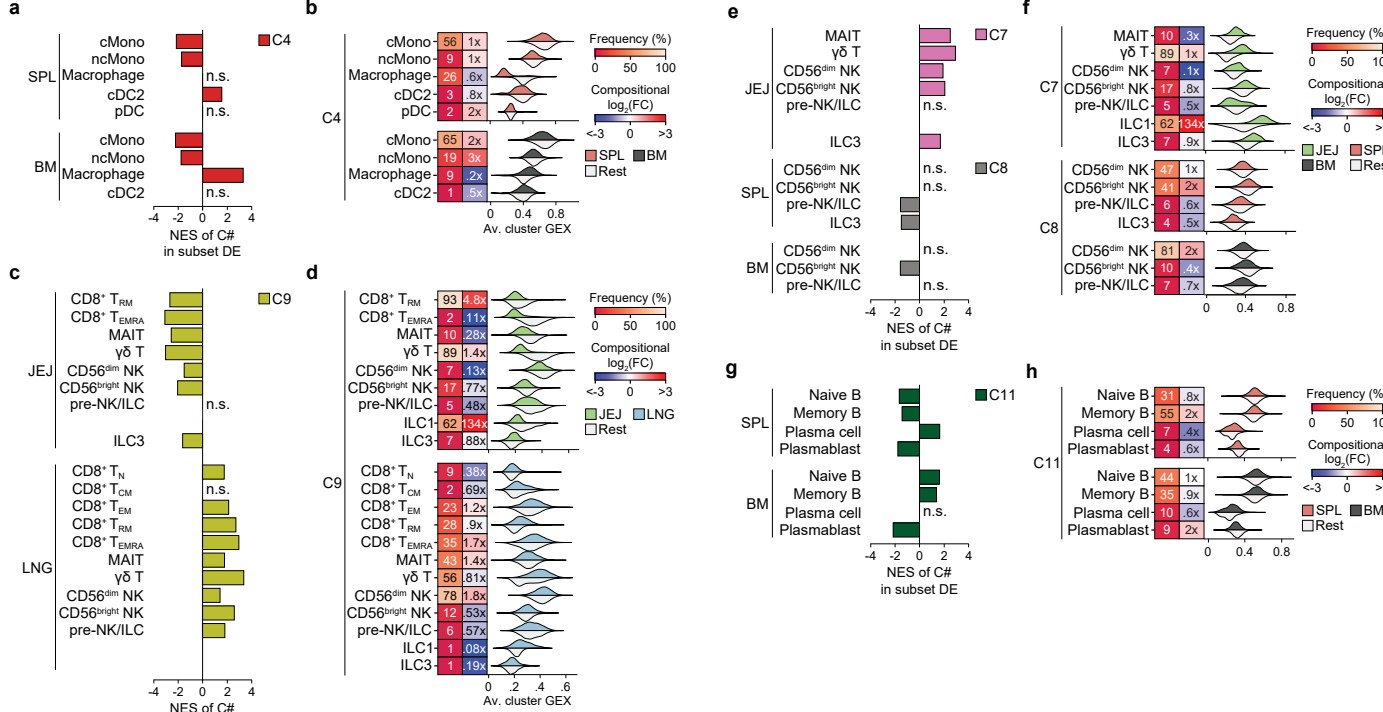

**Extended Data Fig. 3 | Tissue localization shapes gene expression profiles of immune cell subsets. a-h)** Evaluation of additional gene clusters in subset-level DE analysis. Bar plots (**a,c,e,g**) show normalized enrichment scores (NES) of indicated gene clusters in subset-level DE by pre-ranked GSEA. Plotted bars denote adj. p-val (FDR) < 0.05 and n.s. denotes non-significance. Heatmaps (**b,d,f,h**) show subset frequencies (Freq. (%)) as a percentage of their lineage within each tissue, fold changes (FC) in subset frequency in indicated tissue versus other tissues, and split violin plots show average gene cluster expression in indicated tissue versus other tissues.

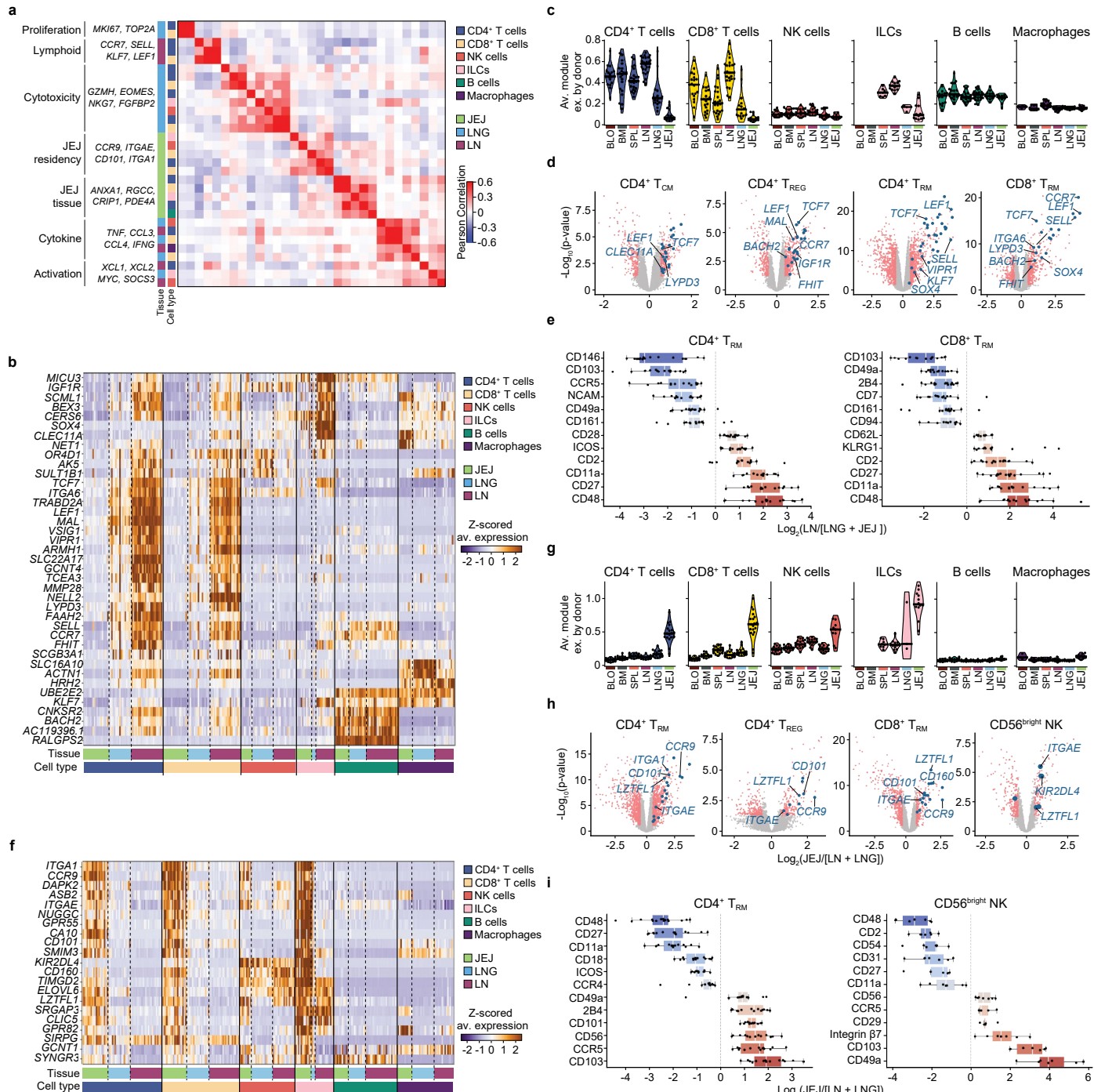

**Extended Data Fig. 4 | Factorization reveals shared tissue gene expression programs across immune lineages. a)** Correlogram of hierarchically clustered Pearson correlations between factor gene scores generated by consensus-scHPF. Cell type models, tissue enrichment, and representative genes per module are indicated. **b-i)** Analysis of the consensus-scHPF Lymphoid module (**b-e**) and JEJ residency module (**f-i**). Heatmaps (**b,f**) show z-scored average expression for each module across immune cell subsets and tissues. Violin plots (**c,g**) show module gene expression across immune subsets and expanded tissue groupings within donors. Volcano plots (**d,h**) show DE genes (adj. p-val (FDR) < 0.05; shown

in pink) from subset-level DE analysis by linear mixed model (dreamlet) in selected immune cell subsets across indicated tissues. Genes from the module are highlighted in blue. Box plots (**e,i**) show differential expression of surface proteins (as LFC) across tissues for selected immune cell subsets. Box plots depicted as median (center), interquartile range (IQR; box), and whiskers extending to 1.5×IQR. Dots represent individual donors with sufficient cell numbers for inclusion. Abbreviations: BLO (blood), BM (bone marrow), SPL (spleen), LN (lymph node), LNG (lung), JEJ (jejunum).

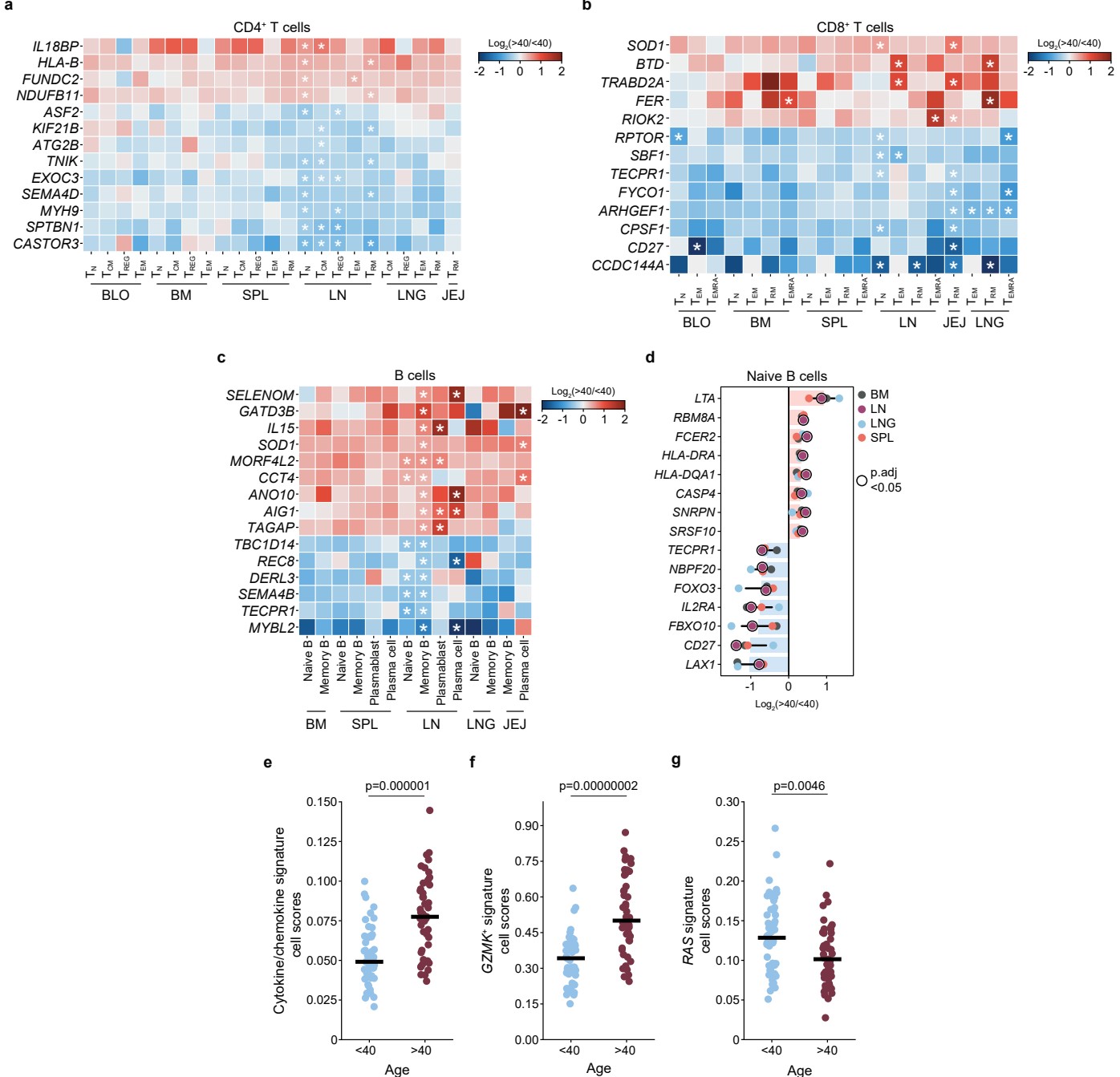

**Extended Data Fig. 5 | Features of immune aging are shared across lymphocyte subsets and validated in an independent cohort. a-c)** Heatmaps of age-associated genes across multiple tissue-subset combinations among CD4+T cells (**a**), CD8+T cells (**b**), or B cells (**c**). Statistical significance determined by linear mixed model (dreamlet). * denotes adj. p-val (FDR) < 0.05. **d)** Bar-and-dot plots of DE genes by age in naïve B cells by subset-level linear mixed model (dreamlet). Bars show median LFC across tissues, error bars represent 95% CI, dot color indicates tissue with significant genes (adj. p-val (FDR) < 0.05) outlined in black. (**e-g**) Dot plots showing factor cell scores from the CD8+T cell Cytokine factor (**e**), CD8+T cell *GZMK*+ signature (**f**), and B cell *RAS*+ signature (**g**) on respective cell types in human blood samples from the Human Immune Health Atlas. Each dot represents a donor with statistical significance determined by two-sided Wilcoxon rank sum test across age.

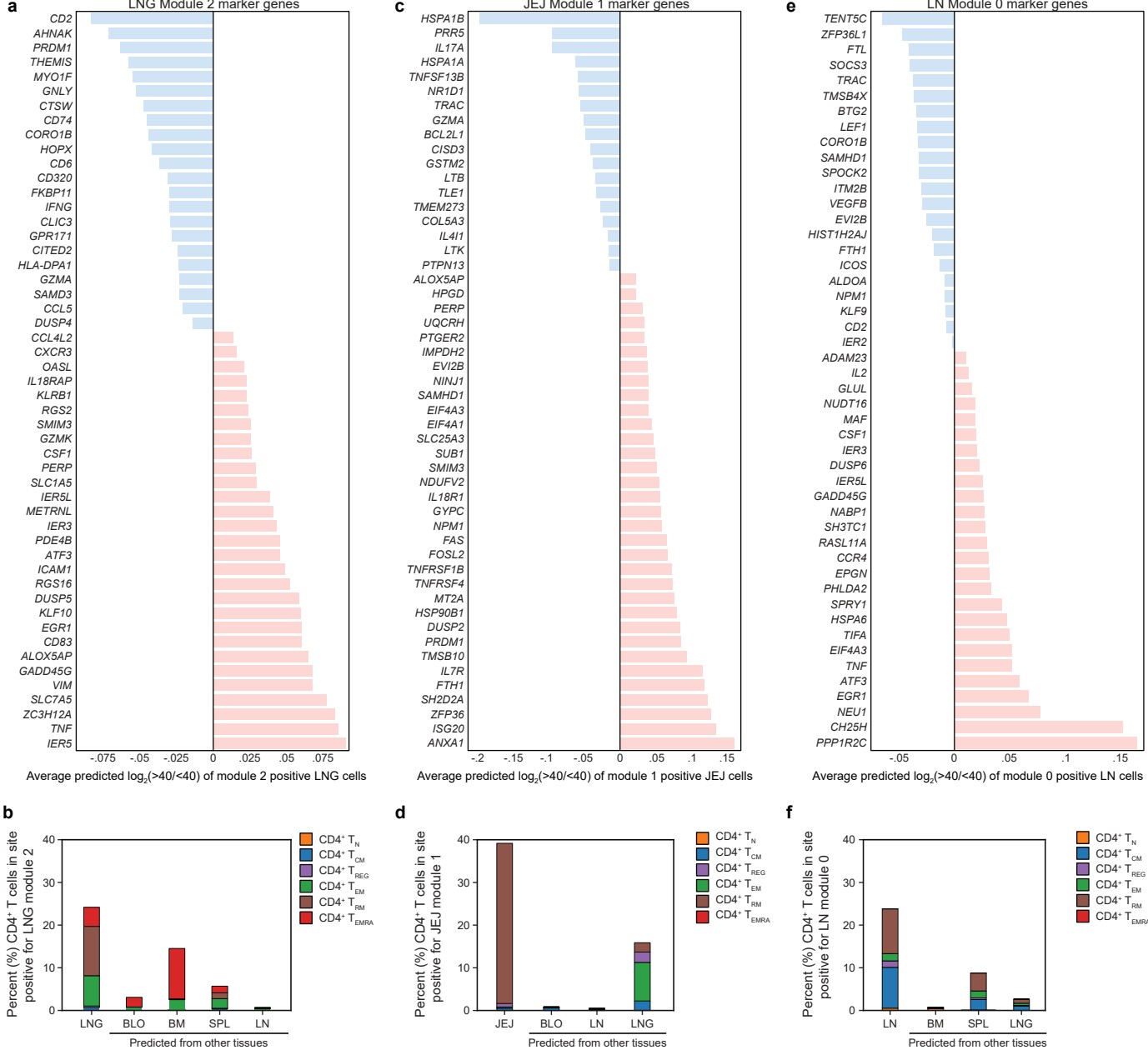

**Extended Data Fig. 6 | MrVI uncovers modules of age-associated changes in CD4⁺ T cells. a-f)** Modules from MrVI analysis of CD4⁺ T cells in the LNG (**a,b**), JEJ (**c,d**) and LN (**e,f**). Bar plots (**a,c,e**) show average predicted LFC ( > 40 y.o. / < 40 y.o.) of top marker genes predicted to change similarly with age in module-positive cells for each tissue model. Stacked bar plots (**b,d,e**) show subset distribution (%) of module-positive cells, colored by subset.

# Reporting Summary

## Statistics

For all statistical analyses, confirm that the following items are present in the figure legend, table legend, main text, or Methods section.

| n/a | Confirmed | |
|---|---|---|
| ☐ | ☒ | The exact sample size ($n$) for each experimental group/condition, given as a discrete number and unit of measurement |
| ☒ | ☐ | A statement on whether measurements were taken from distinct samples or whether the same sample was measured repeatedly |
| ☐ | ☒ | The statistical test(s) used AND whether they are one- or two-sided *Only common tests should be described solely by name; describe more complex techniques in the Methods section.* |
| ☐ | ☒ | A description of all covariates tested |
| ☐ | ☒ | A description of any assumptions or corrections, such as tests of normality and adjustment for multiple comparisons |
| ☐ | ☒ | A full description of the statistical parameters including central tendency (e.g. means) or other basic estimates (e.g. regression coefficient) AND variation (e.g. standard deviation) or associated estimates of uncertainty (e.g. confidence intervals) |
| ☐ | ☒ | For null hypothesis testing, the test statistic (e.g. $F$, $t$, $r$) with confidence intervals, effect sizes, degrees of freedom and $P$ value noted *Give P values as exact values whenever suitable.* |
| ☒ | ☐ | For Bayesian analysis, information on the choice of priors and Markov chain Monte Carlo settings |
| ☐ | ☒ | For hierarchical and complex designs, identification of the appropriate level for tests and full reporting of outcomes |
| ☐ | ☒ | Estimates of effect sizes (e.g. Cohen's $d$, Pearson's $r$), indicating how they were calculated |

*Our web collection on statistics for biologists contains articles on many of the points above.*

## Software and code

Policy information about availability of computer code

| Data collection | Cell Ranger v6.0.0, 10x Genomics |
|---|---|
| Data analysis | Code for data processing and downstream analysis is available at the Github Repository https://github.com/YosefLab/CZI-Immuneaging.<br><br>Scipy v1.6.3<br>decontX v1.7.3<br>scikit-learn  v0.24.2<br>CellTypist v0.9<br>scvi-tools v0.14.5<br>HashSolo v1.7.2<br>Scrublet v1.7.2<br>Scanpy v1.7.2<br>MMocHi v0.2.1<br>Dreamlet v1.4.1<br>scHPF v0.1<br>mrVI v0.3<br>Decoupler v1.3<br>blastn v2.13.0<br>IgBLAST v1.19.0<br>TIgGER v1.0.1<br>Dandelion v0.3.1 |

```
seaborn v0.13.2
statsmodels v0.14.1
gseapy v1.1.3
scCODA v0.1.9
popV v0.5.1
AnnData v0.9.2
```

For manuscripts utilizing custom algorithms or software that are central to the research but not yet described in published literature, software must be made available to editors and reviewers. We strongly encourage code deposition in a community repository (e.g. GitHub). See the Nature Portfolio guidelines for submitting code & software for further information.

## Data

Policy information about availability of data

All manuscripts must include a data availability statement. This statement should provide the following information, where applicable:
- Accession codes, unique identifiers, or web links for publicly available datasets
- A description of any restrictions on data availability
- For clinical datasets or third party data, please ensure that the statement adheres to our policy

The global, T, NK/ILC, B and Myeloid cells h5ad datasets are available at Lattice : Human Cell Atlas https://cellxgene.cziscience.com/collections/cc431242-35ea-41e1-a100-41e0dec2665b. Raw sequencing (Fastq) files are available in Sequence read archive (SRA) under accession SRP559768 and BioProject accession PRJNA1215450. Data are also available in gene expression omnibus under accession number GSE299043.

## Research involving human participants, their data, or biological material

Policy information about studies with human participants or human data. See also policy information about sex, gender (identity/presentation), and sexual orientation and race, ethnicity and racism.

| | |
|---|---|
| Reporting on sex and gender | Sex of human organ donors was provided by organ procurement organization or biorepository and indicated in the manuscript (14 males, 10 females). When appropriate, sex was included as a covariate in the analysis. Gender was not provided or considered in study design. |
| Reporting on race, ethnicity, or other socially relevant groupings | Race, ethnicity, or other socially relevant groupings were not used in the manuscript. |
| Population characteristics | Deceased human organ donors from 20-75 years of age, both male and female, and CMV+ and CMV- serology. All donors were free of cancer, chronic disease, seronegative for hepatitis B, C and HIV and did not show evidence for active infection based on blood, urine, respiratory and radiological surveillance testing. All covariates are noted in the manuscript. |
| Recruitment | Donors were identified through collaborations with the LiveOnNY organ procurement organization or through the Cambridge Biorepository for Translational Medicine. |
| Ethics oversight | Human tissues were obtained from deceased organ donors and therefore not subject to IRB assurances for human subjects protections. Donors from New York, USA were obtained through a materials transfer agreement with LiveOnNY and donors from Cambridge, UK were obtained from the Cambridge Biorepository for Translational Medicine (REC 15/EE/0152). |

Note that full information on the approval of the study protocol must also be provided in the manuscript.

# Field-specific reporting

Please select the one below that is the best fit for your research. If you are not sure, read the appropriate sections before making your selection.

☒ Life sciences    ☐ Behavioural & social sciences    ☐ Ecological, evolutionary & environmental sciences

For a reference copy of the document with all sections, see nature.com/documents/nr-reporting-summary-flat.pdf

# Life sciences study design

All studies must disclose on these points even when the disclosure is negative.

| | |
|---|---|
| Sample size | Data from 24 human organ donors were acquired from all tissues that were available for research purposes. Sample size was determined based on tissue availability and representation of donors > or < 40 years of age. |
| Data exclusions | scRNA-seq from the skin, liver and colon were acquired on select donors and removed from downstream analysis due to low cell or donor number. Non-immune cells were also removed for analysis as the focus of the study was on the immune system. |

| | |
|---|---|
| Replication | Where possible, data from individual donors were treated as independent observations to ensure robustness and reproducibility. This study involves profiling of immune cells from up to 14 sites of 24 individual donors. Replication was based on site and donor. |
| Randomization | All samples acquired from deceased human organ donors were analyzed, no randomization was required. |
| Blinding | This study involves immune cell profiling from multiple tissue and donors isolated directly from live human samples. It is an exploratory study and did not require blinding for outcomes because there were not experiments set up to obtain outcomes. |

# Reporting for specific materials, systems and methods

We require information from authors about some types of materials, experimental systems and methods used in many studies. Here, indicate whether each material, system or method listed is relevant to your study. If you are not sure if a list item applies to your research, read the appropriate section before selecting a response.

## Materials & experimental systems

| n/a | Involved in the study |
|---|---|
| ☐ | ☒ Antibodies |
| ☒ | ☐ Eukaryotic cell lines |
| ☒ | ☐ Palaeontology and archaeology |
| ☒ | ☐ Animals and other organisms |
| ☒ | ☐ Clinical data |
| ☒ | ☐ Dual use research of concern |
| ☒ | ☐ Plants |

## Methods

| n/a | Involved in the study |
|---|---|
| ☒ | ☐ ChIP-seq |
| ☒ | ☐ Flow cytometry |
| ☒ | ☐ MRI-based neuroimaging |

## Antibodies

| | |
|---|---|
| Antibodies used | Biotin anti-human CD235ab Antibody (100µg) (Biolegend, Cat. No.: 306618)<br>Biotin anti-human CD66b Antibody (100µg) (Biolegend, Cat. No.: 305120)<br>Biotin anti-human CD326 (EpCAM) Antibody (100µg) (Biolegend, Cat. No.: 324216)<br>TotalSeq™-A0251 anti-human Hashtag 1 Antibody (BioLegend, Cat. No.: 394601)<br>TotalSeq™-A0252 anti-human Hashtag 2 Antibody (BioLegend, Cat. No.: 394603)<br>TotalSeq™-A0253 anti-human Hashtag 3 Antibody (BioLegend, Cat. No.: 394605)<br>TotalSeq™-A0254 anti-human Hashtag 4 Antibody (BioLegend, Cat. No.: 394607)<br>TotalSeq™-A0255 anti-human Hashtag 5 Antibody (BioLegend, Cat. No.: 394609)<br>TotalSeq™-A0256 anti-human Hashtag 6 Antibody (BioLegend, Cat. No.: 394611)<br>TotalSeq™-A0257 anti-human Hashtag 7 Antibody (BioLegend, Cat. No.: 394613)<br>TotalSeq™-A0258 anti-human Hashtag 8 Antibody (BioLegend, Cat. No.: 394615)<br>TotalSeq™-C0251 anti-human Hashtag 1 Antibody (BioLegend, Cat. No.: 394661)<br>TotalSeq™-C0252 anti-human Hashtag 2 Antibody (BioLegend, Cat. No.: 394663)<br>TotalSeq™-C0253 anti-human Hashtag 3 Antibody (BioLegend, Cat. No.: 394665)<br>TotalSeq™-C0254 anti-human Hashtag 4 Antibody (BioLegend, Cat. No.: 394667)<br>TotalSeq™-C0255 anti-human Hashtag 5 Antibody (BioLegend, Cat. No.: 394669)<br>TotalSeq™-C0256 anti-human Hashtag 6 Antibody (BioLegend, Cat. No.: 394671)<br>TotalSeq™-C0257 anti-human Hashtag 7 Antibody (BioLegend, Cat. No.: 394673)<br>TotalSeq™-C0258 anti-human Hashtag 8 Antibody (BioLegend, Cat. No.: 394675)<br>TotalSeq™-C0259 anti-human Hashtag 9 Antibody (BioLegend, Cat. No.: 394677)<br>TotalSeq™-C0260 anti-human Hashtag 10 Antibody (BioLegend, Cat. No.: 394679)<br>TotalSeq™-A Custom Human Panel (BioLegend, Cat. No.: 99786)<br>TotalSeq™-C Human Universal Cocktail, V1.0 (BioLegend, Cat. No.: 399905) |
| Validation | Antibody cocktails consist of well-validated lyophilized antibodies for cell surface markers at optimized concentrations for single cell sequencing analyses provided by BioLegend and used as directed at one vial per test. All hashtags were used at a 1:100 dilution, all biotinylated antibodies were used at 1:20 dilutions. The manufacturer has validated antibody cocktails on human PBMCs and we have confirmed their effiicacy on human organ donor tissues in pilot experiments and previous studies. Each lot of biotinylated and hashtag antibodies were quality control tested by immunofluorescent staining with flow cytometric analysis and the for the hashtags the oligomer sequence is confirmed by sequencing. |

# Plants

**Seed stocks**
*Report on the source of all seed stocks or other plant material used. If applicable, state the seed stock centre and catalogue number. If plant specimens were collected from the field, describe the collection location, date and sampling procedures.*

**Novel plant genotypes**
*Describe the methods by which all novel plant genotypes were produced. This includes those generated by transgenic approaches, gene editing, chemical/radiation-based mutagenesis and hybridization. For transgenic lines, describe the transformation method, the number of independent lines analyzed and the generation upon which experiments were performed. For gene-edited lines, describe the editor used, the endogenous sequence targeted for editing, the targeting guide RNA sequence (if applicable) and how the editor was applied.*

**Authentication**
*Describe any authentication procedures for each seed stock used or novel genotype generated. Describe any experiments used to assess the effect of a mutation and, where applicable, how potential secondary effects (e.g. second site T-DNA insertions, mosiacism, off-target gene editing) were examined.*

