## [Peer Review File · Nature Immunology]

Multimodal profiling reveals tissue-directed signatures of human immune cells altered with age

Corresponding Author: Dr Donna Farber

Version 0:

Decision Letter:

22nd Jan 2025

Dear Dr. Farber,

Thank you for your response to the referees' comments on your Article, "Multimodal profiling reveals tissue-directed signatures of human immune cells altered with age". Although we are interested in the possibility of publishing your study in Nature Immunology, the remaining issues raised by the referees need to be addressed.

Please revise along the lines specified in your letter. At resubmission, please include a "Response to referees" detailing, point-by-point, how you addressed each referee comment (please specify the page number and the figures where the new data is found). If no action was taken to address a point, you must provide a compelling argument. This response will be sent back to the referees along with the revised manuscript.

Please include a revised version of any required reporting checklist. It will be available to referees to aid in their evaluation. The Reporting Summary can be found here: <https://www.nature.com/documents/nr-reporting-summary.pdf>

When submitting the revised version of your manuscript, please pay close attention to our <https://www.nature.com/nature-portfolio/editorial-policies/image-integrity> Digital Image Integrity Guidelines and to the following points below:

When re-submitting your manuscript, please ensure that any supplementary figures that are more critical to the manuscript's conclusions are converted to Extended Data figures to increase these data's visibility. Please note, Extended Data figures and tables are online-only (appearing in the online PDF and full-text HTML version of the paper), peer-reviewed display items that provide essential background to the Article but are not included in the printed version of the paper due to space constraints or being of interest only to a few specialists. A maximum of ten Extended Data display items (figures and tables) is typically permitted.

Link Redacted

We hope to receive your revised manuscript within two-four weeks. If you cannot send it within this time, please let us know. We will be happy to consider your revision so long as nothing similar has been accepted for publication at Nature Immunology or published elsewhere.

Nature Immunology is committed to improving transparency in authorship. As part of our efforts in this direction, we are now requesting that all authors identified as 'corresponding author' on published papers create and link their Open Researcher and Contributor Identifier (ORCID) with their account on the Manuscript Tracking System (MTS), prior to acceptance. ORCID helps the scientific community achieve unambiguous attribution of all scholarly contributions. You can create and link your ORCID from the home page of the MTS by clicking on 'Modify my Springer Nature account'. For more information please visit www.springernature.com/orcid.

Sincerely,

Ioana Staicu, Ph.D.
Senior Editor
Nature Immunology

Tel: 212-726-9207
Fax: 212-696-9752
www.nature.com/ni

Reviewers' Comments:

Reviewer #1 (Remarks to the Author):

Wells et al. employ multi-modal single-cell sequencing to comprehensively profile human immune cell subsets across various tissue types and age groups in a cohort of 24 organ donors. This valuable resource integrates transcriptomic data, surface proteome analysis (~130 proteins), and TCR/BCR profiling to provide deeper insight into the complexity of the human immune system and will, no doubt, be useful to the broader scientific community. The authors further dive into their resource to identify unique age-related, immune cell transcriptional signatures within and across tissues, revealing new insight into tissue-related aspect of immune aging. While well-written and detailed, there are some issues to be addressed to enhance the clarity and robustness of the authors' conclusions.

While the authors do an excellent job creating detailed and consistent immune cell labeling across tissues, authors take a somewhat indirect two-step computational approach in their transcriptional analyses, which constitutes determining tissue-specific DEGs at a lineage level, then looking for geneset enrichment of unique DE clusters within more defined immune cell subsets. (Fig 3) This type of approach makes it hard to determine whether an immune cell subset has a similar overall expression profile in one tissue vs another. Moreover, it is, as the authors do state, confounded by tissue-specific composition. While the authors do provide some metrics of composition, there are no statistics shown for the NES (Fig 3c, e,g), freq or cluster enrichments, making the conclusion hard to fully interpret.

There are a number of interesting age-related transcriptional findings presented in this paper linked with cytokine secretion and signaling. Although I think confirming all the observations presented within this study is excessive, it is important to show an example of whether these types of transcriptional findings translate into more direct protein responses. To this end, one of the most novel aging findings is the increase in IL-18 - IL18R RNA expression B:T cells in tissues with age. Could these data be confirmed in external datasets and/or at a protein level? This would help significantly bolster the impact of this data resource.

In human studies from blood, donor-specific variation is usually the largest contributor. How much variation is found by donor vs by tissue site?

Chronic CMV infection can have a significant impact on the composition of immune cell in the blood, especially T cells. How does CMV infection impact T cells in other tissue sites? Does it play any role in the GZMK CD8 or CD4 T cell observations seen with age?

Reviewer #2 (Remarks to the Author):

In the revised manuscript, the authors have addressed all of my previous concerns. The computational analysis was substantially expanded to comprehensively characterise differential expression and differential abundance across tissues and age groups at lineage or even subtype resolution. As an alternative to the requested in situ validation, the authors validate their key findings by analysing published datasets for the respective cell populations. Considering the scarcity of donor material, I believe that the provided analyses are sufficient.

With the extended analysis this dataset is ready for publication and represents a valuable resource for the community.

Dominic Grün

Reviewer #3 (Remarks to the Author):

The manuscript by Wells and colleagues arises from a collaborative effort between the University, the Wellcome Sanger Institute, and Columbia University. The primary aim was to explore immune system differences across tissues and age groups. They analyzed RNA and surface protein expression in over 1.25 million immune cells from 24 donors aged 20-75. The authors claim their findings reveal how tissue and age influence immune cell identity. This includes transcriptome, proteome, and TCR/BCR sequencing, as well as the application of advanced analytic tools for single-cell data from cohort studies. The authors also employed automated annotation and statistical models to ensure data accuracy and reproducibility. Nevertheless, the relatively small sample size (24 donors) and the variability in the number of cells and organs analyzed for each donor limit the robustness of the conclusions. As such, the study's findings may not fully characterize the human immune system across different tissues and ages. Without this broader applicability, the study does offer a valuable immune cell resource, but several previous studies (including some by the same authors) have already generated similar data, as noted by the authors themselves and these data can offer only a small increment in the scientific knowledge. The manuscript underwent a revision showing many doubts and concerns from the reviewers. The author responded to the reviewer's comments, not providing additional data. In some cases they made or modify some of the analyses. However, the basic concerns about reproducibility of the data and robustness of the results remain.

Major comments:

1. As reported by the authors the study allowed "unambiguous assessment of tissue effects by intra-donor comparisons". And I agree with this statement reflecting that this study does provide insights for these individuals but the relevance to the human population remains uncertain since its genetic and non-genetic heterogeneity.
2. The study is not original since previously performed similar studies also mentioned by the authors. The study provides technical and conceptual advances, but to my opinion these are minor and not major and, more importantly, not validated by independent methods and on other individuals.
3. The authors agree that there are differences in the cellular composition and immune subset annotation between the dataset used in this study and in the dataset from their previous publication. The authors are unable to prove that this discrepancy is not due to technical protocol or variability in human samples, but propose that also the used annotation itself can generate different results since different annotations has been used in the two studies. To demonstrate this the authors could use the new annotation on the old data and show that the old data with the new annotation provides the same results as the new data.
4. The variability in the number of cells of various lineages across donors and tissues can be a feature of the underlying biology, but also a technical bias due to batch effects. Subsampling is a way to avoid analysis bias, but from the paper I was not able to understand how many donors and tissues have been filtered out. The study already suffers because of a small sample size in comparison to the high variability in human being especially in old age.
- 5-6. In line with the previous point, analyzing 7 "old" individuals cannot provide reliable results on human aging. Under this aspect, the paper, describe the aging of those individuals but I am not convinced that this can be relevant for human population.
7. The authors did not improve the description of author contributions that should be more complete and report each single contribution for each experiment/analysis.

Version 1:

Decision Letter:

Our ref: NI-A38357A

30th Apr 2025

Dear Dr. Farber,

Thank you for submitting your revised manuscript "Multimodal profiling reveals tissue-directed signatures of human immune cells altered with age" (NI-A38357A). It has now been seen by one of the original referees and their comments are below. We are happy to inform you that if you revise your manuscript appropriately according to our editorial requirements, your manuscript should be publishable in Nature Immunology.

I will now pre-edit the current version of your paper. We will also perform detailed checks on your paper and will send you a checklist detailing our editorial and formatting requirements in about two weeks. Please do not upload the final materials and make any revisions until you receive this additional information from us.

While waiting for the pre-edit check, please deposit all omic and code data into public repositories so that the accession codes are readily available to be added in the revised manuscript. We cannot accept the paper without the codes.

In addition, all corresponding authors need to update and link their ORCID to their Nature account. We cannot accept the paper without this information. We suggest that you look into this while waiting for the pre-edited manuscript as well. Should you have any query or comments about ORCID, please do not hesitate to contact our editorial assistant at immunology@us.nature.com.

If you had not uploaded a Word file for the current version of the manuscript, we will need one before beginning the editing process; please email that to immunology@us.nature.com at your earliest convenience.

Thank you again for your interest in Nature Immunology. Please do not hesitate to contact me if you have any questions.

Sincerely,

Ioana Staicu, Ph.D.
Senior Editor
Nature Immunology

Tel: 212-726-9207
Fax: 212-696-9752
www.nature.com/ni

Reviewer #1 (Remarks to the Author):

I have re-read all the reviewers comments and authors edits/responses. I feel that the work contained in this revised manuscript definitely extends much beyond their previous work and helps move the field of immunology and immune aging forward - from looking mainly in the blood into now what is actually going on in the tissues. This type of human work is extremely challenging, costly and time-intensive to set-up, execute and sustain. In my opinion, reviewer 3 comments about small numbers not translating to entire human populations across age is valid but also, could be stated for almost every study on humans to-date. The methodology and rigor that authors used to address the known fact that there is variation in human sampling and aging is more important and it is done well.

Point by Point Response to Reviewers

“Multimodal profiling reveals tissue-directed signatures of human immune cells altered with age”

Manuscript number: NI-A38357-T

Corresponding authors: Donna L. Farber, Nir Yosef, Peter Sims, Joanne Jones

We appreciate the careful evaluation of our manuscript by the reviewers and have fully addressed the comments and concerns by additional revisions to the manuscript. We performed new analyses to address the major comments concerning robustness and generalizability of our results by validating our findings using a newly available human aging dataset from peripheral blood, assessed specific effects of CMV infection on age-related signatures, clarified our methodology regarding tissue signatures, and emphasized the many novel aspects of our study and findings. The new analyses are presented in **Figure 5, Extended Data Figure 5, Figure 6, Supplementary Figures 9 and 10**, and additional Supplementary tables along with corresponding revisions throughout the text, figure legends, and methods, indicated by underlining in the uploaded files. Enumerated below is a point-by-point response to each reviewer comment including a description of the new data and/or analyses and the corresponding revisions to the manuscript. The reviewers' comments are in *italics*, and our responses are in plain font.

Reviewer #1:

Wells et al. employ multi-modal single-cell sequencing to comprehensively profile human immune cell subsets across various tissue types and age groups in a cohort of 24 organ donors. This valuable resource integrates transcriptomic data, surface proteome analysis (~130 proteins), and TCR/BCR profiling to provide deeper insight into the complexity of the human immune system and will, no doubt, be useful to the broader scientific community. The authors further dive into their resource to identify unique age-related, immune cell transcriptional signatures within and across tissues, revealing new insight into tissue-related aspect of immune aging. While well-written and detailed, there are some issues to be addressed to enhance the clarity and robustness of the authors' conclusions.

Response: We appreciate this favorable evaluation and have directly addressed the remaining issues of clarity and robustness as detailed below.

1. While the authors do an excellent job creating detailed and consistent immune cell labeling across tissues, authors take a somewhat indirect two-step computational approach in their transcriptional analyses, which constitutes determining tissue-specific DEGs at a lineage level, then looking for geneset enrichment of unique DE clusters within more defined immune cell subsets. (Fig 3) This type of approach makes it hard to determine whether an immune cell subset has a similar overall expression profile in one tissue vs another. Moreover, it is, as the authors do state, confounded by tissue-specific composition. While the authors do provide some metrics of composition, there are no statistics shown for the NES (Fig 3c, e,g), freq or cluster enrichments, making the conclusion hard to fully interpret.

Response: Our two-phase computational approach to identify tissue-specific signatures and their expression in specific subsets of T, B, innate, and myeloid-lineage cells is presented in **Figure 3** and **Extended Data Figure 3**. While our method was designed to address challenges posed by smaller sample sizes and mitigate the impact of tissue-specific composition, we acknowledge that it may obscure direct comparisons of immune cell subset expression profiles across tissues. We therefore included direct comparison of gene and surface protein expression for the major resident immune cells present in sufficient frequencies across sites (CD4 and CD8 TRM, plasma cells, and macrophages) between different tissues (intestine, lungs, lymphoid organs) in **Figure 4**. This analysis yielded comprehensive site-specific signatures for resident immune cells across all the different compartments examined, as mentioned by the reviewer.

Regarding GSEA statistics in **Figure 3** and **Extended Data Figure 3**, we only displayed normalized enrichment scores (NES) for cases where enrichment or depletion was statistically significant. The “n.s.” designation (not significant) was included for a small number of tests to indicate non-significant results. We have clarified this point in the legend for **Figure 3** and **Extended Data Figure 3** and added **Supplementary Tables 7-8** to provide GSEA statistics.

2. *There are a number of interesting age-related transcriptional findings presented in this paper linked with cytokine secretion and signaling. Although I think confirming all the observations presented within this study is excessive, it is important to show an example of whether these types of transcriptional findings translate into more direct protein responses. To this end, one of the most novel aging findings is the increase in IL-18 - IL18R RNA expression B:T cells in tissues with age. Could these data be confirmed in external datasets and/or at a protein level? This would help significantly bolster the impact of this data resource.*

Response: Although our multimodal dataset of major immune cell lineages and subsets across blood, lymphoid organs, and mucosal tissues is unique to this study and will serve as an important reference dataset for the field, we identified three scRNAseq datasets with significant representation of individuals across age for validation of specific signatures: 1. A human lung atlas¹ which contained immune cells; 2. A bone marrow atlas²; and 3. A newly available multimodal PBMC data from the Sound Life cohort (25-65yrs, n=96) included as part of the Human Immune Health Atlas³, that matched the age range of our donors. For a given aging signature, we selected the validation data based on the abundance of each cell type in each tissue. For example, macrophages are most abundant in the lung, and so we used the lung atlas for validating our macrophage aging signature (rather than PBMC or bone marrow atlases). Using this strategy, we included additional validations, covering all the scHPF-based gene signatures that are discussed in the paper.

Specifically, the previous version of our manuscript used the lung atlas to validate the macrophage APOE/TREM2 aging signature (**Figure 5k**) and the CD8 T cell cytokine signature (**Figure 6e**) and the bone marrow atlas to validate the B cell RAS signaling aging signature (**Figure 6o**). The *GZMK* signature was also previously identified as being age-associated in mouse splenic and human blood T cells⁴. In the revised manuscript, we further validated both of the CD8 T cell aging signatures (Cytokines and *GZMK*) and the B cell RAS aging signature using the Human Immune Health Atlas PBMC dataset and these results are presented in **Extended Data Figure 5e-g** and presented in the results (pp. 18-19). We also validated increased expression of IL-18-associated genes (*IL-18R* and *IL18RAP*) in this same health atlas PBMC dataset (**Supplementary Figure 9**). These validation results show that our aging analysis identified signatures that are seen in a larger cohort of individuals across a similar age range.

These additional validations further bolster the relevance and the impact of our findings, demonstrating consistent age-related transcriptomic changes across independent datasets.

3. *In human studies from blood, donor-specific variation is usually the largest contributor. How much variation is found by donor vs by tissue site?*

Response: To assess the relative contributions of donor- vs. tissue-level variation, we performed variance decomposition analysis across major immune lineages presented in **Figure 5a** of the revised manuscript (these data were originally in Extended Data Fig. 5a but we moved to a main figure given their importance). Notably, tissue consistently emerges as the dominant

source of variation, surpassing donor-level covariates (e.g., age, sex, and CMV status) or technical covariates (e.g., 10x chemistry). While donor-specific variation typically dominates single-site studies (e.g., blood), our multi-tissue analysis reveals tissue as the primary driver of variation—a key finding of our work. To further enumerate the contribution of co-variables to our aging analysis, we present a variance composition analysis for the top genes that vary with age for each lineage in **Figure 5b** (described in the results section, p. 15). Many of these age-associated genes also vary by tissue. Together, these results emphasize the major impact of tissue on immune cell identity and its changes over age.

4. Chronic CMV infection can have a significant impact on the composition of immune cell in the blood, especially T cells. How does CMV infection impact T cells in other tissue sites? Does it play any role in the GZMK CD8 or CD4 T cell observations seen with age?

Response: We included CMV serostatus as a co-variate in all of aging analyses, including the covariate-aware differential expression analysis using Dreamlet (**Figure 5**) and scHPF factorization analysis (**Figures 5, 6**), given the known effect of CMV in exacerbating age-associated changes in immune cells. We previously reported increased TEMRA accumulation with age in blood, spleen and lungs in CMV seropositive compared to seronegative donors^{5,6}. In the revised manuscript, we compared the composition of the major immune lineages (myeloid, T, B and innate cells) in the 16 CMV+ and 8 CMV- donors, identifying significant differences in splenic immune cell subsets (macrophages, NK cells), monocytes in bone marrow, and increased trends for TEMRA with age across multiple sites (**Supplementary Fig. 10a**).

To interrogate whether the aging transcriptional signatures identified here were differentially associated with CMV infection status, we performed scHPF analysis within each major lineage of our cohort (see **Figure 1b** for CMV infection status for each donor). We found that both the age-associated GZMK CD8 T cell signature and a new CD8 signature #4 were statistically associated with CMV+ donors (**Supplementary Fig. 10c-f**). The new CD8 signature #4 is marked by cytolytic mediators (*GNLY*, *GZMH*) and homing molecule (*CX3CR1*). Together, these findings indicate certain effects of CMV infection on age-associated changes in immune cells, specifically on myeloid cell composition and increased cytolytic functions for CD8 T cells. These new analyses and results are described in the revised manuscript (pp. 19-20).

Reviewer #2

In the revised manuscript, the authors have addressed all of my previous concerns. The computational analysis was substantially expanded to comprehensively characterise differential expression and differential abundance across tissues and age groups at lineage or even subtype resolution. As an alternative to the requested in situ validation, the authors validate their key findings by analysing published datasets for the respective cell populations. Considering the scarcity of donor material, I believe that the provided analyses are sufficient.

With the extended analysis this dataset is ready for publication and represents a valuable resource for the community.

Dominic Grün

Response: We appreciate the favorable review and acknowledgement of the revised analysis and importance of the dataset.

Reviewer #3

The manuscript by Wells and colleagues arises from a collaborative effort between the University, the Wellcome Sanger Institute, and Columbia University. The primary aim was to explore immune system differences across tissues and age groups. They analyzed RNA and surface protein expression in over 1.25 million immune cells from 24 donors aged 20-75. The authors claim their findings reveal how tissue and age influence immune cell identity. This includes transcriptome, proteome, and TCR/BCR sequencing, as well as the application of advanced analytic tools for single-cell data from cohort studies. The authors also employed automated annotation and statistical models to ensure data accuracy and reproducibility. Nevertheless, the relatively small sample size (24 donors) and the variability in the number of cells and organs analyzed for each donor limit the robustness of the conclusions.

As such, the study's findings may not fully characterize the human immune system across different tissues and ages. Without this broader applicability, the study does offer a valuable immune cell resource, but several previous studies (including some by the same authors) have already generated similar data, as noted by the authors themselves and these data can offer only a small increment in the scientific knowledge. The manuscript underwent a revision showing many doubts and concerns from the reviewers. The author responded to the reviewer's comments, not providing additional data. In some cases they made or modify some of the analyses. However, the basic concerns about reproducibility of the data and robustness of the results remain.

Major comments:

1. As reported by the authors the study allowed "unambiguous assessment of tissue effects by intra-donor comparisons". And I agree with this statement reflecting that this study does provide insights for these individuals but the relevance to the human population remains uncertain since its genetic and non-genetic heterogeneity.

Response: While the respective part of our study focuses on intra-donor comparisons to unambiguously assess tissue effects, we acknowledge the importance of addressing genetic and non-genetic heterogeneity across the human population. To rigorously account for donor-level variability in the comparisons we have made between tissues, we encoded the donor identity as a random effect in our statistical models, ensuring that tissue effects were assessed while controlling for individual differences. Additionally, we employed a conservative estimate of effects by adjusting for measurable, donor-level covariates (e.g., sex, age, CMV status). The results shown are therefore conservative estimates of tissue-specific effects, requiring consistency across donors of different ages, gender, and other key covariates.

Regarding reproducibility or relevance to the human population, we also note that all age-associated gene signatures identified with scHPF and discussed in our study were validated in independent cohorts. This includes validation of the macrophage *APOE/TREM2* and CD8 cytokine signatures in the lung atlas (**Figure 5k**), validation of both CD8 T cell aging signatures (Cytokines and *GZMK*) and the B cell RAS signaling signature in a newly available multimodal dataset of human PBMC (the Human Immune Health Atlas; added in this revision) (**Extended Data Figure 5e-g**), and additional validation of the B cell aging signature in a bone marrow atlas (**Figure 6o**) (See also response #2 to Reviewer #1). Importantly, all three of these validation datasets, while focused on specific tissue sites, include larger cohorts than the current study (lung atlas n=34, bone marrow atlas n=39, PBMC Human Immune Health Atlas n=96). Together, these validations underscore the robustness and generalizability of our findings.

Furthermore, our laboratory (Farber) has extensively profiled immune cells from over 700 organ donors for the past 15 years, using diverse methodologies, including flow cytometry, functional assays, population and single cell transcriptomic (see reviews⁶⁻⁸). These studies consistently demonstrate robust and reproducible tissue-specific effects across donors. For example, we previously identified tissue-specific profiles for TRM cells including part of the lymphoid signature in lymph node CD8⁺T cells marked by increased TCF-1 expression⁹, gut-specific profiles across multiple platforms¹⁰⁻¹², and distinct lung T cell profiles compared to gut and lymphoid organs¹⁰. Similarly, our analysis of NK cells revealed tissue-specific subset compositions and gut-specific profiles that remain stable with age¹³. These findings align with the tissue-specific adaptations which are identified and expanded upon in this study, particularly in the gut, further validating the relevance of our results. We have discussed how our current results for tissue-specific effects align, are supported, and/or suggested by previous studies using different approaches in the revised discussion (pp.22-23).

In summary, by employing conservative and rigorous statistical approaches we ensured a robust assessment of the tissues effects. Combined with validation across independent cohorts and our extensive prior work, these findings strongly support the applicability of our results to the broader human population.

2. The study is not original since previously performed similar studies also mentioned by the authors. The study provides technical and conceptual advances, but to my opinion these are minor and not major and, more importantly, not validated by independent methods and on other individuals.

Response: We respectfully disagree with the assertion that this study lacks originality or major conceptual advances. The previous scRNAseq study of tissue immune cells¹⁴ was presented in the context of a new computational cell classifier and was limited in scale and scope—being derived from fewer donors (with 70% of the data from just two individuals) and lacking the depth to address key biological questions such as tissue- and age-specific effects. By contrast, our study profiles >1.2 million cells from 24 donors across six decades of life, employing CITE-seq (as opposed to RNA-seq only) to simultaneously analyze the transcriptome and >130 surface proteins across 10 tissue sites. This multimodal approach, combined with our newly developed MMoCHi classifier (recently published and validated¹⁵) and the mrVI method for integration and counterfactual analysis (to appear in Nature Methods), enables rigorous annotation of cell subsets (for which the protein data was key, see Supplementary Fig. 3 and question #3), and integrated analysis of tissue- and age-mediated effects across immune lineages and subsets.

Our findings reveal major new insights, including tissue-specific signatures linked to migration, residence, function, and metabolism (**Figs. 3-4, Extended data Figs. 3-4**) as well as new and independently validated age-associated changes in immune cell subsets that are also related to tissue (**Figs. 5-7, Extended data Fig. 5**). Between sites, we identified stem-like features in lymph node T cells, distinct residency and functional profiles in gut vs. lung lymphocytes, and distinct tissue-specific profiles for macrophages and plasma cells. Our age analysis revealed age-associated reductions in macrophage function via the *APOE/TREM2* pathway, functional signatures for CD8 T cells including an age-associated GZMK signature that differed by site and subset, and a reduction in RAS signaling in memory B cells with age in lymphoid organs. These results were obtained using multiple statistical approaches (differential expression, gene set enrichment, factorization, and multivariate modeling), and combined with validation across independent cohorts to ensure robustness and reproducibility.

We are confident that this comprehensive, multimodal dataset—spanning 10 tissues, 24 donors, and six decades of life—will serve as the definitive reference for understanding human tissue immunity and will drive future discoveries in the field.

3. The authors agree that there are differences in the cellular composition and immune subset annotation between the dataset used in this study and in the dataset from their previous publication. The authors are unable to prove that this discrepancy is not due to technical protocol or variability in human samples, but propose that also the used annotation itself can generate different results since different annotations has been used in the two studies. To demonstrate this the authors could use the new annotation on the old data and show that the old data with the new annotation provides the same results as the new data.

Response: First, the previous study did not have sufficient number of donors, cells (with 70% of the data derived from 2 donors), nor the age range to analyze aging, which is a major emphasis of the present study. Our study is unique in the analysis of age-associated changes across multiple immune lineages in blood, lymphoid organs, and mucosal sites. This point is also stated above and mentioned many times in the manuscript, including the title. So, it is not possible to compare any results from the aging analysis for this dataset and the previous one.

Second, the original study introduced CellTypist as a tool for immune cell annotation using scRNAseq data alone and did not investigate tissue-specific adaptations or shared immune lineage effects, which are central to our current work. By contrast, the present dataset has multimodal profiling integrating scRNAseq and surface protein data using a new multi-modal classifier specifically developed for this study and extensively validated and described in a recent publication¹⁵. This classifier outperformed CellTypist and other classifiers in immune cell annotation and many key immune cell subsets analyzed here (including TEM and TEMRA, naïve and TCM, and NK and $\gamma\delta$ T cells) require surface proteins for accurate annotation¹⁵. For example, we performed a direct comparison of MMoCHi and CellTypist using CITE-seq and FACS on the same PBMC samples showing that MMoCHi achieves 92% agreement with FACS-based annotations, while CellTypist agreement was much lower—at 50%. In addition, we have compared MMoCHi and CellTypist annotations within our current dataset, showing improved accuracy across subsets and lineages with the MMoCHi-based annotations (**Supplementary Fig. 3**). This evidence underscores the improved accuracy of our multimodal annotation approach.

In summary, our current study provides novel insights into tissue-specific adaptations and age-related changes in immune cells, supported by a robust, multimodal dataset and rigorous statistical analysis.

4. The variability in the number of cells of various lineages across donors and tissues can be a feature of the underlying biology, but also a technical bias due to batch effects. Subsampling is a way to avoid analysis bias, but from the paper I was not able to understand how many donors and tissues have been filtered out. The study already suffers because of a small sample size in comparison to the high variability in human being especially in old age.

Response: To address concerns regarding variability in cell numbers across donors and tissues, we employed rigorous analytical approaches to ensure robust and interpretable results. First, all differential expression analyses (**Figs. 3-6**) and supporting analyses (**Fig. 7**) were performed using Dreamlet, a pseudobulk method that is less susceptible to variations in cell numbers, except in cases of extremely small sample sizes. Prior to analysis, we therefore filtered out data points (cell type x tissue x donor) with insufficient cell counts to ensure reliable results (see

Methods). To improve transparency, added tables in the revised manuscript listing the data points included in each analysis (see Supplementary Tables 5, 8 and 15).

Second, we clarify that our subsampling and downsampling approaches were designed to mitigate representation bias without reducing sample size (i.e. number of cell type x tissue x donor combinations). For the scHPF modeling, we subsampled cell numbers and downsampled molecular counts to ensure balanced representation across donors and tissues. For example, when comparing CD4+ T cells across donors, we ensured all donors were represented by the same number of cells in the model. Importantly, this approach does not remove entire samples (e.g., tissues or donors), preserving our sample size with minimizing bias (see Methods).

5-6. *In line with the previous point, analyzing 7 "old" individuals cannot provide reliable results on human aging. Under this aspect, the paper, describe the aging of those individuals but I am not convinced that this can be relevant for human population.*

Response: Our cohort of 24 donors ranging 20-75yrs comprises ten younger (<40yrs) and fourteen older donors (>40yrs) (see **Fig. 1b**, Methods). Comparisons between younger and older donors includes this delineation of young or older than 40yrs, as stated in the legends of all figures related to the aging analysis (**Figures, 5-7; Extended data figure 5, 6 and Supplementary figures 8-10**). Regarding relevance of this dataset of human immune cells from blood and 9 tissue sites of 24 human donors to the human population, please see the response to comment #1 of this reviewer regarding the validation of our aging signatures to independent datasets from human blood, bone marrow, and lungs. Currently, there are no other comprehensive datasets for human immune cell aging from different tissues, and our dataset will be a key reference dataset in this regard.

Therefore, our robust statistical approaches, validated findings, and the known tissue-driven effects on immune composition we have observed in flow cytometry in previous studies provide strong evidence that the data presented here are representative of the human population.

7. *The authors did not improve the description of author contributions that should be more complete and report each single contribution for each experiment/analysis.*

Response: We indicated author contributions based on the journal guidelines.

References:

1. Natri, H. M. *et al.* Cell type-specific and disease-associated eQTL in the human lung. *bioRxiv* (2023) doi:10.1101/2023.03.17.533161.
2. Lee, N. Y. S., Li, M., Ang, K. S. & Chen, J. Establishing a human bone marrow single cell reference atlas to study ageing and diseases. *Front. Immunol.* **14**, 1127879 (2023).
3. Gong, Q. *et al.* Longitudinal multi-omic immune profiling reveals age-related immune cell dynamics in healthy adults. *bioRxiv* (2024) doi:10.1101/2024.09.10.612119.
4. Mogilenko, D. A. *et al.* Comprehensive Profiling of an Aging Immune System Reveals Clonal GZMK(+) CD8(+) T Cells as Conserved Hallmark of Inflammaging. *Immunity* **54**, 99-115 e12 (2021).
5. Gordon, C. L. *et al.* Tissue reservoirs of antiviral T cell immunity in persistent human CMV infection. *J. Exp. Med.* **214**, 651–667 (2017).
6. Kumar, B. V., Connors, T. J. & Farber, D. L. Human T Cell Development, Localization, and Function throughout Life. *Immunity* **48**, 202–213 (2018).

7. Gray, J. I. & Farber, D. L. Tissue-Resident Immune Cells in Humans. *Annu. Rev. Immunol.* **40**, 195–220 (2022).
8. Lam, N., Lee, Y. & Farber, D. L. A guide to adaptive immune memory. *Nat. Rev. Immunol.* **24**, 810–829 (2024).
9. Miron, M. *et al.* Human Lymph Nodes Maintain TCF-1(hi) Memory T Cells with High Functional Potential and Clonal Diversity throughout Life. *J. Immunol.* **201**, 2132–2140 (2018).
10. Poon, M. M. L. *et al.* Tissue adaptation and clonal segregation of human memory T cells in barrier sites. *Nat. Immunol.* **24**, 309–319 (2023).
11. Connors, T. J. *et al.* Site-specific development and progressive maturation of human tissue-resident memory T cells over infancy and childhood. *Immunity* **56**, 1894-1909 e5 (2023).
12. Szabo, P. A. *et al.* Single-cell transcriptomics of human T cells reveals tissue and activation signatures in health and disease. *Nat. Commun.* **10**, 4706 (2019).
13. Dogra, P. *et al.* Tissue Determinants of Human NK Cell Development, Function, and Residence. *Cell* **180**, 749-763 e13 (2020).
14. Dominguez Conde, C. *et al.* Cross-tissue immune cell analysis reveals tissue-specific features in humans. *Science* **376**, eabl5197 (2022).
15. Caron, D. P. *et al.* Multimodal hierarchical classification of CITE-seq data delineates immune cell states across lineages and tissues. *Cell Rep Methods* (2025) doi:110.1016/j.crmeth.2024.100938.